# Can Enterprises in China Achieve Sustainable Development through Green Investment?

**DOI:** 10.3390/ijerph20031787

**Published:** 2023-01-18

**Authors:** Sisi Zheng, Shanyue Jin

**Affiliations:** College of Business, Gachon University, Seongnam 13120, Republic of Korea

**Keywords:** executives’ overseas experience, government environmental subsidies, green investment, investor attention, sustainable development

## Abstract

Enterprises have emerged as primary actors in environmental protection owing to the increasingly severe global energy crisis and environmental pollution. Companies can reduce operational costs, achieve environmental social responsibility, and enhance their green image by increasing their green investments. Simultaneously, companies can gain support from investors, governments, and other stakeholders for improving their sustainable development. This study uses fixed-effects regression models to analyze the impact of green investment on corporate sustainability in Chinese listed companies for the period from 2010 to 2020. It also investigates the moderating effects of government environmental subsidies, investor attention, and executives’ overseas experience on the relationship between green investment and corporate sustainability. The data used in this study were not only obtained from the China Stock Market & Accounting Research (CSMAR) database but also collected manually from the annual reports and social responsibility reports of listed companies using web crawler technology. And the robustness test was conducted by removing the epidemic year and replacing the range of independent variables and 2SLs. This study uses Stata 17.0 to filter and process the data. The results show that green investment can significantly improve the sustainability of enterprises; besides, government subsidies, investors’ attention, and executives’ overseas experience all play a positive role in moderating the positive effect of green investment on the sustainable development of enterprises. Further analysis of this study finds that the moderating effect is more significant in non-state-owned enterprises and highly polluting enterprises. This study contributes to broadening the theory related to the green development of enterprises and environmental governance and provides theoretical support for enterprises to make green investment decisions and green transformations.

## 1. Introduction

In the last few decades, socially responsible investment (SRI) has gained increasing importance, accounting for approximately 10% of all investment money [1]. This figure increased by 324% between 1995 and 2007 [2]. Comparisons of the returns of ethical and conventional investments have been used by many academics to study a variety of topics [3,4,5], including the substance and motivation of SRI [6] and the effect of socially responsible investment on business sustainability [7]. Green investment, as a subset of SRI, is defined as a necessary investment for significant energy conservation, emission reduction, and energy consumption reduction; it is conducive to creating a green environment [8] and works as a catalyst for promoting the establishment of a clean environment for the next generation [9].

The United Nations convened the Sustainable Development Summit in September 2015, which adopted “Transforming our World: The 2030 Agenda for Sustainable Development” and codified the Sustainable Development Goals (SDGs) agreed upon by all member states. Taking into account the 3 dimensions of sustainable development (economic, social, and environmental), the Sustainable Development Goals (SDGs) comprise 17 primary goals and 169 secondary goals. For instance, promoting sustained, inclusive, and sustainable economic growth; inclusive and sustainable industrialization and innovation; addressing the climate crisis caused by greenhouse gas emissions; protecting, restoring, and promoting sustainable ecosystems; reducing energy consumption; and adopting affordable and clean energy. All European Union (EU) member states are committed to meeting the Sustainable Development Goals by 2030, and the transition to a green economy has emerged as a worldwide economic development trend. By signing this pact, EU member states voluntarily commit to coordinating their economic, social, and environmental development to reduce their environmental impact. China, a key player in the sustainable development of the global economy, society, and environment, strives to practice green development while simultaneously accelerating its economic growth.

Since its accession to the World Trade Organization, China has made world-renowned economic achievements, but because of a lack of strict environmental restrictions and inexpensive production, it experiences serious environmental pollution [10]. In recent years, the Chinese government has emphasized the infrastructure development of ecological civilization and green development, committing to the promotion of green practices among enterprises through energy conservation and emission reduction [11]. Green investments, such as energy savings and carbon reduction, have become crucial considerations for enterprises [11]. According to corporate social responsibility (CSR) theory, enterprises should work toward achieving profit goals while fulfilling their social responsibilities to protect the environment and use resources wisely [12]. The theory also states that environmental social responsibility leads to long-term positive development [12].

The Chinese government has used administrative regulations, economic instruments, and market mechanisms with some success to limit pollutant emissions [13], but if serious problems such as resource scarcity, environmental pollution, and ecological damage are not fundamentally addressed, China will struggle to achieve its green and sustainable development goals [14]. Environmental equity, according to Li and other scholars, is essential for achieving resource use efficiency and sustainable development, while local government environmental governance capacity and corporate green innovation technologies can effectively mitigate environmental inequality [15]. According to Ren et al., green investment can reduce environmental pollution by enhancing energy conservation and emission reduction capacities, expanding technological innovation capacities, and modernizing industrial structures [13]. Green energy consumption and investment have a small but positive impact on economic growth, according to Balcilar and other macroeconomists [16]. However, some academics are optimistic about the impact of green investment on economic development. The literature has evaluated the influence of green investment on high-quality economic development and carbon emission reduction, concluding that green investment has become crucial in promoting high-quality economic development [17] and reducing carbon emissions [18].

Stakeholders such as governments, investors, and executives also play a pivotal role in green investment and the sustainability of enterprises. For instance, Voica et al. contend that government involvement in green investments could be significant. Governments can create institutional and legislative frameworks to support green infrastructure investments [19] in order to create a green economy and advance sustainable development. Dobler et al. claimed that investors evaluate regulatory, reputational, physical danger, lawsuit, and supply chain risks when making portfolio selections and prefer to invest in enterprises with green concepts and good sustainability [20]. Labor economics considers intelligence one of the drivers of endogenous company growth, which has led to an increasing number of companies bringing in talented individuals with experience in overseas activities and board business decisions [21]. Studies have found that chief executive officers (CEOs) with advanced overseas education have high environmental awareness and are willing to implement green management for corporate sustainability [22]. Slater and Heather [23] argue that CEOs with overseas experience influence the corporate environmental performance of a company and that CEOs’ overseas posts lead to improved corporate social performance. As Mariassunta and Andrei [24] explain in their study of the impact of overseas experience on the performance of firms in emerging markets, the hiring of executives with international experience is beneficial to a company’s long-term viability. According to their findings, firm performance significantly improves when directors with international experience are hired.

Based on the analysis of the literature and the current research context, we focus not only on the impact of green investment on energy savings, emission reduction, and environmental management, but also on how microenterprises can achieve their own sustainable development goals through green investment and then promote the realization of a virtuous cycle of corporate environmental spending and sustainable development. To fill this research gap, this study constructs a study of Chinese listed companies in the Shanghai and Shenzhen stock markets from 2010 to 2020. The impact of the green investment behavior of Chinese firms on improving corporate sustainability is empirically examined, and the results show that green investment by Chinese firms can contribute to enhancing corporate sustainability. This study further explores the mechanism of the interaction between the two using the moderating effect. The empirical study finds that government environmental subsidies, investor attention, and executive overseas experience can positively moderate the contribution of green investment to corporate sustainability. It provides an important theoretical reference for enterprises to increase green investment and achieve sustainable development goals. It also provides strong theoretical support and policy guidance for regulators, investors, and executives on how to help enterprises achieve sound economic and environmental sustainability.

In addition, most of the existing studies focus on the impact of environmental investment on corporate financial performance and environmental performance from the micro level; the impact of market incentive-based environmental regulations, etc. on corporate sustainability; the impact of green investment on environmental sustainability; and the impact of the green economy on sustainable development from the macro perspective using provincial panel data. Few papers link the green investment behavior of microfirms with their sustainability and further verify the mechanism of action of the relationship based on the stakeholder perspective. This study provides an important theoretical reference for expanding the paradigm of corporate green investment research and fills this gap. This study analyzes the impact of corporate green investment behavior on corporate sustainability from three stakeholder perspectives that influence corporate investment behavior, such as the government, investors, and executives. It reveals an important mechanism of action of green investment behavior affecting corporate sustainable development, enriches the literature perspective of corporate green investment and sustainable development, and can provide a reference for enterprises to achieve sustainable development goals. The purpose of the current study is to examine whether corporate green investments are truly effective in improving corporate sustainability and to evaluate the impact of stakeholders on the relationship between the two.

This study expands the theory surrounding the green development of firms and environmental governance and offers enterprises theoretical support for making green investment decisions and undertaking a green transformation.

## 2. Theoretical Background and Hypothesis Testing

### 2.1. Green Investment and Corporate Sustainability

Studies based on neoclassical economic theory argue that corporate environmental protection expenditures crowd out productive capital, leading to a decrease in productivity and profit [25]. Green investment by businesses is a “passive activity” that complies with governments’ environmental rules [26]. This characterization is due to the high barrier to entry for businesses looking to invest in environmental safeguards, such as the purchase of environmental protection equipment and funding of green technology research and development (R&D), and the low probability of a positive return [27]. According to studies based on the Porter hypothesis, environmental investments raise expenses in the short term but have the potential to spur technical innovation and generate “innovation compensation” over the long term, thereby improving productivity and performance [28]. Meanwhile, green investment has also been regarded as an “active behavior” of enterprises. On the one hand, businesses benefit from increasing the rate at which they use their resources, attracting more customers and other sources of funding, building a positive reputation among their target audiences, and achieving sustainable development [29]. Taylor et al. concluded that upholding environmental responsibility may result in a short-term loss of economic benefits but guarantees long-term sustainable development [30]. On the other hand, businesses can boost customer loyalty and reputation by engaging in green practices, thus reaping economic benefits from an indirect approach [31].

According to the theory of environmental economics, balancing and coordinating the economy and the environment is important owing to their direct impact on each other [32]. The theory emphasizes the importance of considering the relationship between economic development and the environment, coordinating the relationship between people and nature, and maintaining ecological balance as the foundation of enterprise development while addressing the growing material needs of people [32]. According to the conclusions of a large number of studies, one of the key financial grounds for sustainable growth is an investment in environmentally friendly industries [33,34,35,36,37]. Economic, social, and environmental well-being are the three pillars of what Elkington calls “the triple bottom line” of business sustainability. Elkington contended that businesses should not prioritize economic efficiency over social and environmental benefits and should keep all three bottom lines in mind [38]. For enterprises, sustainability is concerned with future capacity building and capital accumulation, and it should be a collection of behaviors; that is, while continuously pursuing economic benefits, enterprises should ensure that they minimize damage to social and environmental interests [39]. The green investment consists mostly of capital investments in energy-saving and renewable technologies [17]. Green investment, as a new approach to enterprise resource allocation, can allocate limited resources to green technology, energy savings, emission reduction, renewable resource development, etc., to reduce energy consumption and improve resource utilization while achieving pollution control. According to Brown et al., government taxes encourage businesses that produce pollution to invest in adopting and using cleaner production methods. They contend that taxing pollutant emissions encourages businesses to invest more in green technologies to switch to cleaner production. Green investments, they claim, have higher marginal returns for polluting businesses [40] and are more likely to encourage long-term corporate growth. In addition, Alam et al. investigated the relationship between corporate green R&D expenditure and environmental performance in important developed nations and found that enterprises’ use of their own resources is conducive to improving environmental performance and achieving sustainable competitiveness [41]. Zheng et al. concluded that firms could invest in environmental protection as a social investment and that green investment in environmental protection could improve the efficiency and sustainability of firms [42]. Empirical evidence from Sun et al. supports the notion that green investments and clean energy both play significant roles in reducing environmental pollution and fostering environmentally sustainable development [43]. Xin et al. found that CSR can promote high-quality corporate development through improved green innovation, environmental investment, and corporate governance [44]. On the basis of the above analysis, this study proposes the following hypothesis:

**Hypothesis** **1:**
*Firms’ green investments help to improve their sustainability.*


### 2.2. The Regulating Role of Government Environmental Subsidies

In recent years, the Chinese government has pledged support for green growth. Green investment, as a key tool for constructing a resource-conserving and environmentally friendly society, can foster the long-term growth of China’s natural capital, industrial output, and standard of living. Green investment generally entails a large investment amount and is characterized as having a slow effect, high riskiness, and a low short-term return [45]. Such characteristics limit enterprises’ willingness to make green investments. Environmental protection subsidies, one of the most important vehicles of a government’s market-based environmental control [46], exert a considerable influence on the production and management decisions of businesses. First, the government has boosted infrastructure construction through environmental subsidies in order to provide firms with adequate hardware support facilities for green innovation and other investment activities. This decreases the market risk associated with green innovation and increases the predictability of green technology product returns. Second, the local government’s environmental expenditure communicates to the market the government’s determination to carry out ecological and environmental protection and stimulates economic entities to improve environmental protection awareness, which on the one hand is conducive to directing social capital and high-quality talents to green fields such as ecology, energy conservation, environmental protection, and environmental governance, and on the other hand is conducive to fostering environmental governance. This will contribute to the expansion of green technology innovation. Thirdly, the government encourages businesses to adopt green investing practices in order to lower tax burdens by implementing environmental regulatory measures such as environmental levies. From a resource-based perspective, environmental protection subsidies can impact the green investment behavior of private enterprises through a mechanism for the direct replenishment of green investment resources. Blazsek et al. found that green R&D subsidies can promote green patent applications in firms [47]. Liu and other scholars concluded from empirical analysis that environmental taxes have a catalytic effect on firms to increase their environmental investments and that environmental taxes promote firm performance by increasing environmental investments [48]. Hattori found that green subsidies can address the reduction in firms’ green R&D investments owing to green technology spillovers [49]. That is because environmental subsidies will stimulate businesses to cut production costs and boost productivity through breakthroughs in green technology, which will directly raise their profit margins and entice additional businesses to follow suit [50]. Continuous innovation and widespread adoption of green technologies enable sustainable economic and environmental growth. According to resource dependency theory, businesses must collect resources from the surrounding social environment in order to thrive. The stakeholder theory states that businesses should do everything possible to satisfy the demands and expectations of various stakeholders. As Freeman [51] explained, internal stakeholders consist of employees, managers, and owners, while external stakeholders include customers, regulators, and the community at large. Freeman believed that accomplishing goals in collaboration with these social parties would be simple if the links were tight between them. From these two hypotheses, we can deduce that the Chinese government has a significant impact on business competition [52], that private companies in China receive environmental subsidies in the form of financial aid, and that the government has high hopes for the sector’s future green growth. Moreover, business organizations will raise their green investment budgets and maximize the environmental subsidy funds to satisfy government mandates. Some academics support the “Porter hypothesis” [53], which contends that effective environmental regulations can encourage businesses to offset the costs of environmental protection through green innovation and enhance their market profitability and product quality. Such regulations could give domestic businesses a competitive edge in the global marketplace while potentially increasing industrial productivity. The “crowding-in effect” of green investment is made advantageous by government subsidies for environmental protection. On the one hand, they can compensate for the lack of resources and external risks caused by enterprises’ green investment under environmental regulations, alleviate the fund gap in the process of green investment [54], and have a positive incentive effect on enterprises’ green investment. On the other hand, according to signaling theory, businesses can project their green brand image to the general public and signal to stakeholders a positive relationship between the government and the organization by obtaining environmental protection subsidies and making green investments. From an investor perspective, investors tend to build investment relationships with enterprises that have good government resources; that is, they prefer enterprises that can receive government subsidies. Moreover, companies are willing to make green investments to win diversified support [52]. The favorable signals to the outside are also convenient for enterprises to obtain venture capital and bank credit [55], stabilize the source of funds for green R&D, and create synergy with environmental regulations, thus further promoting the development of green investment and its sustainability. Using empirical analysis, Liu et al. came to the conclusion that environmental taxes improve firms’ performance by spurring increased environmental investment and that environmental taxes have a catalytic effect on this increase in investment [48]. Government environmental spending, according to Hong et al., can lead to businesses adopting a free-rider mentality and avoiding environmental responsibility, which has a detrimental effect on how effective that spending is. Corporate green investment is negatively impacted by government environmental spending, or “crowding out” [56]. However, Zhang et al. came to the conclusion that government subsidies have a favorable impact on corporate environmental investment [57] and that the association between government subsidies and corporate environmental investment can help decision-makers develop policies and allocate limited resources. The following hypotheses are proposed on the basis of the foregoing analysis.

**Hypothesis** **2:**
*Government environmental subsidies have a positive moderating effect on green investment for improving corporate sustainability.*


### 2.3. Moderating Effect of Investor Attention

Referring to stakeholder theory, Rhenman proposed a comprehensive definition: “Stakeholders are interdependent with the company, relying on it to achieve their personal goals while also supporting the company’s survival.” Thus, many different types of investors care about environmental quality and utilize their influence to affect corporate business decisions through yearly shareholder meetings [58], thereby creating a constraint mechanism between investors’ attention and company actions. Deng et al. discovered that the occurrence of environmental events may strengthen firms’ environmental investments and that extremely high investor attention leads to higher returns for green firms [59]. This, in turn, may lead firms to increase their green investments because of the attention that investors pay to green companies. In fact, investors weigh potential dangers such as regulatory, reputational, physical, litigation, and supply chain risks when making portfolio selections. To protect their wealth rather than the environment for future generations, investors who view environmental contamination and environmental laws as potential financial hazards are increasingly turning to green investments [9,20]. Today, more than half of all institutional investors publicly commit to responsible investing [60]. For these investors (and the companies whose operations they fund), sustainability also means self-preservation [61]. Investors’ focus on corporate sustainability as a result of corporate green investments is consistent with the universal owner theory, which states that large institutional investors benefit from promoting sustainability because their universal portfolios expose them to the broad environmental costs faced by individual companies whose securities they hold [62]. In addition to internal pressures on firms to improve sustainability through green investments, external pressures originate from stakeholders such as investors. For example, investors may view a low ranking on environmental metrics as a financial risk either because it indicates inefficient and wasteful underlying company operations or because it signals potential legal exposure to future tort litigation or increased regulatory scrutiny [63]. Although the link between sustainability and financial performance is not entirely clear, a growing number of mainstream investors want great consistency between the portfolios they hold and their value [63]. Othar Kordsachia and colleagues argued that the propensity for sustainability held by institutional investors is also a driving force behind firms’ green investments. They argue that companies with more sustainable institutional investor ownership have higher carbon risk awareness and that sustainable institutional ownership is positively related to firms’ environmental performance [64]. In addition, they claim that firms with more sustainable institutional investor ownership have higher environmental performance. According to the findings of Pan et al.’s research, public concern for the environment has a positive impact on the effectiveness of green investments made by businesses [65]. Thus, this study proposes the following hypotheses on the basis of the previous analysis:

**Hypothesis** **3:**
*Investors’ attention has a positive moderating effect on green investments for improving corporate sustainability.*


### 2.4. Moderating Effect of Executives’ Overseas Experience

Behavioral finance theory states that under environmental uncertainty, people’s decision-making behavior is affected by personal psychological factors and changes in the social environment; such contention is not completely rational. Risk attitudes and decisions often deviate from the optimal model of financial theory and exhibit diversity and variability [66]. Moreover, people’s values and cognitive preferences shift over time [66], and their upbringing and work history are important factors in their decision-making. Executives with overseas experience have received advanced education, exposure to cutting-edge business knowledge, and professional training abroad and thus have a broad perspective and rich social resources. According to Hambrick and Mason’s (1984) upper-echelon theory, executives play a crucial role in the day-to-day administration of corporations because of their extensive knowledge and field experience. Cao et al. argued that executives with overseas experience have substantial beneficial effects on the behavioral decisions of companies’ green investment and environmental social responsibility [67], and how to make use of executives with overseas backgrounds to improve the efficiency of enterprises’ green investment becomes an important issue. Cui et al. discovered that, when comparing the roles of executives with overseas backgrounds in high and low carbon firms, executives with overseas experience play a larger role in green innovation in high carbon firms and are more conducive to meeting carbon reduction goals [68] in response to social environment needs [69]. Luudgren et al. argued that the environmental preferences of various stakeholders influence firms’ decisions to make green investments [70]. As stakeholders, CEOs who received higher education overseas have high environmental awareness and are willing to implement green management for corporate sustainability [22]. Lopatta et al. concluded that the overseas backgrounds of CEO-led companies provide higher quality sustainability reports [71], suggesting that executives with overseas backgrounds have higher ethics, integrity, and focus on sustainability. With regard to CSR, the concepts and ethical standards of CEOs with international experience tend to be robust [72]. Such CEOs also tend to develop a worldview that encourages CSR initiatives and places value on social responsibility. Compared with their counterparts, companies managed by boards or executives with international backgrounds are more likely to adopt the practices and characteristics of foreign frameworks for corporate governance [73], are less inclined to evade taxes [74], are more efficient in terms of investment [75], and are less likely to be at risk of collapse [76]. Returning executives urge their businesses to increase CSR spending through approaches such as green innovation investments [77,78]. Thus, executives with overseas experience are likely to promote green investments because they have enhanced environmental ethics and care about corporate sustainability. Heavily polluting businesses find difficulties in their operations owing to the increasing public interest in ecological civilization and green development. Executives with overseas experience have strong environmental awareness and the ability to improve green investment efficiency. Based on the reputation effect and regulatory supervision, executives with overseas experience can encourage enterprises to be willing to accept environmental protection behavior, actively meet their legitimacy, consider the interests of stakeholders, optimize resource allocation efficiency, and improve the value realization of green investment. Chen et al. concluded that the overseas background of executives has a significant positive impact on the green technology innovation of enterprises, and that this impact is more obvious in high pollution cities [79]. Therefore, this study proposes the following hypotheses on the basis of the analysis:

**Hypothesis** **4:**
*Executives’ overseas experience has a positive moderating effect on green investments for improving corporate sustainability.*


The theoretical framework of this study is depicted in Figure 1.

## 3. Research Methodology and Design

### 3.1. Data and Samples

The Guide to Environmental Information Disclosure of Listed Companies, published by the former Ministry of Environmental Protection in 2010, proposed to further improve the transparency of corporate environmental information disclosure. On the one hand, considering the Chinese government’s further emphasis on Chinese companies’ participation in environmental protection after 2010, and on the other hand, considering the poor availability of data on corporate green investment before 2010, we set the sample period as 2010–2020. The authors firstly selected all listed companies in the A-share market of the Shanghai and Shenzhen Stock Exchanges from 2010 to 2020 through the China Stock Market & Accounting Research (CSMAR) database as the research sample. Second, enterprises with special treatment, special treatment*, or special transfer status were removed from the sample. Third, the samples that lacked critical information were discarded. Fourth, because the lagged period data are used for the independent variables in this study, the effective sample size is 14,779 after excluding the final sample that did not actually participate in the regression; Fifth, to reduce the effect of outliers, the variables were winsorized at the upper and lower 1% levels (excluding the dummy variables); sixth, we logged the continuous variables to reduce the interference of heteroskedasticity. This study uses Stata 17.0 to filter and process the data, and the results of the statistical analysis were obtained using Stata 17.0.

The data on green investment of listed companies in Shanghai and Shenzhen A-shares in China from 2010–2020 were obtained mainly by downloading annual reports of enterprises, information from the websites of listed companies, social responsibility reports of listed companies, and manually collating capital expenditure and expensed expenditure of enterprises on environmental protection and summing them up by using web crawler technology. The information on government funding for environmental protection was compiled from various sources, including annual company reports, social responsibility reports, company websites, and environmental department websites, using web crawling techniques and manually collating the aggregated results. The investor focus indicators were taken from the Google Search Volume Index (GSVI). Executives’ overseas experience was gleaned from the biographical information in the CSMAR database, which, if necessary, was supplemented with the manually collated biographies of executives disclosed in annual reports. All other required public company data was obtained from the CSMAR database.

### 3.2. Definition and Measurement of Variables

#### 3.2.1. Explained Variables

The explanatory variable in this study is corporate sustainability, which is defined as a company’s ability to remain profitable and experience high growth despite intense competition and shifting market conditions. The static model of Van Horne, as cited by Liao et al. [80], is utilized in this study to measure the sustainability of a corporation in terms of its profitability and competitiveness.

#### 3.2.2. Explanatory Variables

According to Eyraud et al. [17], “green investment” is capital spending aimed at reducing greenhouse gas emissions and atmospheric pollution and includes financial investments in renewable technologies, the choice of energy-efficient technologies, and R&D of green technologies. Drawing on Wang et al. [11], this study uses firms’ environmental expenditures to represent green investments. The amount of green investment capitalized and the amount expensed make up the total green investment. The total environmental investment is divided by the total assets at the conclusion of the period to eliminate the scale impact [81]. Due to the small amount of data for this indicator after de-scaling, the variable was multiplied by 100 to enhance readability.

#### 3.2.3. Regulating Variables

In the current work, we refer to the studies of Xinfeng Jiang et al. [82] and Qingyuan Li et al. [83], where government environmental subsidies are calculated using a standardized approach to operating income. Given the small amount of data for this indicator after de-scaling, the variable is multiplied by 100 to enhance readability.

This study employs the GSVI with stock symbols as keywords to gauge investors’ focus on companies. The selection of the GSVI is driven by its popularity as a measure of investor focus in the behavioral finance literature [84,85,86].

In this study, we refer to the studies of Yuan and Wen [87] and Liu et al. [88]. We quantify executives’ international exposure by calculating the percentage of total executives who have worked abroad.

#### 3.2.4. Control Variables

Along with the results of Jin et al. [89], we use firm size (SIZE), asset-liability ratio (LEV), profitability (ROA), cash flow ratio (CFLOW), net asset turnover ratio (ATO), firm growth capacity (GROWTH), market value (TOBINQ), nature of equity (SOE), and age (AGE) as control variables to rule out the effects of any outside influences. We incorporate industry impact (INDUSTRY), a dummy variable with a value of 1 if the firm belongs to the industry and 0 otherwise, to account for the considerable variation in green investment levels across different business sectors. We introduce a year effect (YEAR) to set a dummy variable with a value of 1 if it belongs to the corresponding year and a value of 0 otherwise. This variable accounts for the fact that the level of green investment may fluctuate substantially from year to year because of changes in the macro environment and policies. Table 1 presents the labels and explanations of the study variables.

### 3.3. Model Design

The following model is built on the basis of the study of Jin et al. [89] to examine the effect of green investment on company sustainability:(1)$$SGR_{it}=\alpha_{0}+\alpha_{1}GI_{i,t-1}+\Sigma Control_{it}+{µ}_{i}+\gamma_{t}+\varepsilon_{it}.\;$$

In the regression, the subscripts *i* and *t* in Equation (1) denote individual firms and years, respectively. The explained variable is the sustainability of the firm (SGR). The explanatory variable is a green investment (GI). Green investment is treated with a one-period lag ($G_{i,t-1}$) to account for the lagged effect of green investment (capitalized and expensed expenditures) that contributes to improving the firm’s sustainability and prevents endogenous and reverse causality difficulties. *Control* refers to the set of control variables that affect sustainability. Further, this study considers industry-fixed effects in addition to individual effects ${µ}_{i}$ and year-fixed effects $\gamma_{t}$, where *ε* is the random error term of the model, to mitigate the impact of individual heterogeneity and year features on company sustainability. Given the possible heteroscedasticity problem, robust standard error regressions are conducted.

To further verify the mechanism of the influence of government environmental subsidies, investor attention, and executives’ overseas experience on green investment to improve corporate sustainability, the interaction terms of moderating variables with green investment are added to the baseline regression model [90]. Hayes [91] suggested regenerating the turnover multiplier term after centering the variables on making the regression equation’s coefficients increasingly informative. Additionally, robust standard errors are used to regress the three moderating variables in light of the potential heteroskedasticity issue. The following model is constructed:(2)$$SGR_{it}=\beta_{0}+\beta_{1}GI_{i,t-1}+\beta_{2}ENVSUB_{it}+\beta_{3}GI_{i,t-1}*ENVSUB_{it}\;+\Sigma Control_{it}+{µ}_{i}+\gamma_{t}+\varepsilon_{it}.$$

As shown in model (2), on the basis of model (1), government environmental subsidies ($ENVSUB_{it}$) and their interaction term ($GI_{i,t-1}*ENBSUB_{it}$) with the lagged-period green investment *GI* were added. In testing model (2), if the coefficient $\beta_{3}$ of the interaction term ($GI_{i,t-1}*ENBSUB_{it}$) is positive and statistically significant, it suggests that government environmental subsidies (*ENVSUB*) can enhance the positive effect of green investment on supporting the sustainable growth of firms.
(3)$$SGR_{it}=\phi_{0}+\phi_{1}GI_{i,t-1}+\phi_{2}ATTEN_{it}+\phi_{3}GI_{i,t-1}*ATTEN_{it}\;+\Sigma Control_{it}+{µ}_{i}+\gamma_{t}+\varepsilon_{it}.\;$$

Model (3) demonstrates that investor attention ($ATTEN_{it}$) and its interaction term ($GI_{i,t-1}*ATTEN_{it}$*)* with the lag period of green investment G*I* are added to the foundation of model (1). If the coefficient $\phi_{3}$ of the interaction term $GI_{i,t-1}*ATTEN_{it}$ in the model (3) is positive and statistically significant, then investor attention (ATTEN) favorably enhances the positive effect of green investment on boosting corporate sustainability.
(4)$$SGR_{it}=\gamma_{0}+\gamma_{1}GI_{i,t-1}+\gamma_{2}OVERSEA_{it}+\gamma_{3}GI_{i,t-1}*OVERSEA_{it}\;+\Sigma Control_{it}+{µ}_{i}+\gamma_{t}+\varepsilon_{i}t.\;$$

Model (4) shows that executives’ overseas experience ($OVERSEA_{it}$*)* and its interaction term ($GI_{i,t-1}*OVERSEA_{it}$*)* with lagged one-period green investment GI are added to the foundation of model (1). If the coefficient $\gamma_{3}$ of the interaction term $GI_{i,t-1}*OVERSEA_{it}$ in the model (4) is positive and statistically significant, then it indicates that executives’ overseas experience (*OVERSEA*) can positively moderate the positive effect of green investment on promoting corporate sustainability.

## 4. Results of the Empirical Analysis

### 4.1. Descriptive Statistics and Correlations

From Table 2, it can be seen that we screened a total sample size of 14,779 for 10 years, and the specific distribution of samples participating in the statistics for each year is shown in Table 2. Due to the one-period data that was used for the independent variables, there was no real sample size for the regression in 2010.

The descriptive data are shown in Table 3. Corporate sustainability (SGR) has a mean value of 0.031, a median value of 0.008, a standard deviation of 0.078, a maximum value of 0.633, and a minimum value of −0.057. As the median is lower than the mean and corporate sustainability is more concentrated at a low level, we can infer that considerable variances exist in the sample’s overall level of corporate sustainability. The sample green investment data follow a normal distribution, and the overall green investment of firms is at a low level, as indicated by the mean value of green investment (*GI*) of 0.045 and standard deviation of 0.005. The mean value of the variable observing executives’ overseas experience (OVERSEA) is 0.071, with a standard deviation of 0.082. The mean value of the variable observing investor attention (ATTEN) is 1.090, with a standard deviation of 0.558. The mean value of the variable observing government environmental subsidies (ENVSUB) is 0.046, with a standard deviation of 0.016. These results show significant differences in the sample companies’ individual characteristics. According to the aforementioned descriptive statistics, the variables chosen for this study have a suitable range of values, and no notable outliers or indicators contradict the regression hypotheses. Moreover, the chosen sample meets the study’s criteria.

Table 4 shows the Pearson correlation coefficients obtained to check for multicollinearity before analyzing the impact of green investment on company sustainability. This table shows a significant connection between GI and SGR of 0.174, indicating that the prior hypothesis is valid. When performing regression analysis, the preferable approach is to have variables that are reasonably independent of one another and where multicollinearity is not a concern, as shown by a correlation analysis in which all correlation coefficients are less than 0.8. Furthermore, the variance inflation factor values of all variables are less than 3, which suggests that the impact of multicollinearity on the primary results can be disregarded.

### 4.2. Analysis of Empirical Results

#### 4.2.1. Green Investment and Corporate Sustainability

In Hausman’s test results, *p* = 0.000; thus, we could account for year effects, sector fixed effects, and individual fixed effects. In our study, we decided to use a fixed-effects regression model. Table 5 presents the results of the green investment and corporate sustainability benchmark regression.

The first column in Table 5 shows that the 1% significance level for the regression coefficient of ${\mathrm{GI}}_{-1}$ is positive, suggesting that increasing enterprises’ green investment significantly enhances their sustainability; hence, H1 is supported.

#### 4.2.2. Moderating Effect of Government Environmental Subsidies on Corporate Sustainability

Government environmental subsidies moderate the relationship between green investment and company sustainability, as shown in the regression results in Column (2) of Table 5. The green investment ($GI_{-1}$*)* coefficient of the model (2) is 1.142, and the interaction term between green investment and government environmental subsidies ($GI_{-1}*ENVSUB$) coefficient is 0.013, both of which are statistically significant at the 1% level. The moderating effect of government environmental subsidies on green investment for business sustainability is strong, as shown by the significant increase in the adjusted R^2^ of the model (4) from 0.197 in the model (1) to 0.261 after adding the interaction term. Hence, H2 is supported.

#### 4.2.3. Moderating Effect of Investor Attention on Corporate Sustainability

The regression results for the moderating impact of investor attention on the ability of green investment to promote business sustainability are shown in Column (3) of Table 5. According to the table, model (3)’s coefficient for green investment ($GI_{-1}$) is 1.186, and the coefficient for the interaction term between green investment and investor attention ($GI_{-1}*ATTEN$) is 0.008, both of which are statistically significant at the 1% level. The moderating effect of investor attention on green investment for corporate sustainability is significant, as indicated by the increase in adjusted R^2^ from 0.197 in the model (1) to 0.222 in the model (3) after adding the interaction variable. Thus, H3 is supported.

#### 4.2.4. Moderating Effect of Executives’ Overseas Experience on Corporate Sustainability

Executives with overseas experience moderate the effect of green investment on company sustainability, as seen in the regression results in Column (4) of Table 5. The table shows that model (4)’s coefficient for green investment ($GI_{-1}$*)* is 0.984, while the coefficient for the interaction term between green investment and executives’ overseas experience (${\mathrm{GI}}_{-1}*\mathrm{OVERSEA}$) is 0.012. Both the coefficients are statistically significant at the 1% level. The moderating effect of executives’ overseas experience on green investment for business sustainability is significant, as indicated by the increase in adjusted R^2^ from 0.197 in the model (1) to 0.250 in the model (4). Hence, H4 is supported.

### 4.3. Robustness Tests

This study adjusts the sample interval and range of independent variables and performs two-stage least squares (2SLS) regression to examine the robustness of the regression of the main effects and thereby further validate the findings.

#### 4.3.1. Tests Based on Sample Interval Adjustment

In 2019, the COVID-19 pandemic swept the globe, profoundly affecting economies, societies, and financial systems worldwide. The introduction of stringent laws to address this pandemic has severely limited corporate activity in many areas. A local trend of reverse globalization has also emerged as a result of the pandemic’s effects, and the effects of the pandemic and the aforementioned trend have paved the way for the notion of sustainable development to become widely accepted worldwide. Opportunities for green investment, such as low-carbon emission-related businesses and the growth of a climate-resilient economy, have emerged as a result of new patterns and trends in green investment and green financing. In 2020, a pivotal year following the COVID-19 outbreak, notable variations in the growth of production, operations, and green investment among businesses were observed. Therefore, we perform robustness tests in this study by omitting the sample firms in 2020 to ensure the accuracy of the aforementioned empirical analysis.

Table 6 shows the results of the regression analysis of green investment for corporate sustainable development after removing the sample of companies from 2020. The regression coefficient of green investment for corporate sustainable development is 1.738 and thus passes the significance test, as shown in Column (1) of the main model results in Table 6. This outcome is consistent with the results of the previous regression analysis and suggests that green investment has a significant promotion effect on corporate sustainable development. After including the moderating variables in Columns (2)–(4), the regression results show that the coefficients of the three interaction terms are 0.013, 0.008, and 0.012, and that all three are statistically significant. When the moderating factors are included, the corrected R^2^ value increases dramatically. Consistent with the findings of prior regression analyses, the results of this study suggest that government environmental subsidies, investor attention, and executives’ overseas experiences positively moderate the impact of green investment on business sustainability.

#### 4.3.2. Tests Based on Range Adjustment of Independent Variables

To expand the sample size of the study, we obtain the data on green investment from the total capitalized and expensed expenditures related to the green environment found in CSR reports and other documents and notes to financial statements. The use of environmental capital expenditures disclosed in “construction in progress” in social responsibility and financial reports to represent green investments is a common approach in academia [92]. To confirm the validity of the aforementioned empirical analysis, this study conducts a second robustness test on the sample using green capital expenditure as the independent variable.

Table 7 displays the findings of the regression analysis conducted on green investment for corporate sustainable development; here, green capital expenditure with a one-period lag serves as the independent variable. Consistent with the findings of prior regression analysis, the results of the main model in Column (1) show that green investment has a considerable promotion effect on corporate sustainable growth, with the regression coefficient being 1.833. With the addition of the moderating factors to the regression through Columns (2)–(4), the coefficients of the three interaction terms are 0.011, 0.007, and 0.010. These values all pass the significance test. When the moderating factors are included, the corrected R^2^ value increases dramatically. Consistent with the findings of earlier regression analysis, the results show that government environmental subsidies, investor attention, and executives’ overseas experiences all positively moderate green investment for corporate sustainability within the sample.

#### 4.3.3. Testing Based on Two-Stage Least Squares

Owing to the diverse selection of control variables, omitted variables, and causality in this study, endogeneity difficulties may occur when benchmark regressions are used to analyze the influence of green investment on business sustainability. Research has shown that the growth of green investments can significantly enhance business sustainability, but this improvement may also affect the level of green investments. This study uses one-period lagged green investment as the explanatory variable in the fixed effects regression model to overcome the reverse causality issue. However, additional endogeneity testing of the model is required to strengthen the validity of the results. This study uses the one-period lagged data of the mean of green investment in the same year in the industry in which the firm is located as an instrumental variable for green investment (GImean1), and a two-stage estimation method is adopted to test it. In this regard, we draw on the study by Fisman et al. [93] and consider that economic activities, such as investment decisions, are easily influenced by the same activities of other firms in the same industry [94]. The same test for the existence of weak instrumental factors and the identifiability of instrumental variables is used while performing the 2SLS regression.

The outcomes of the 2SLS analysis are shown in Table 8. The endogeneity of the variables is addressed, and the results show that the regression coefficient of GI_−1_ on corporate sustainability is 4.426 at the 1% significance level, demonstrating that the growth of green investment plays a substantial role in promoting corporate sustainability. The trustworthiness of the results is further supported by the fact that the baseline regression results, including the instrumental factors, are generally consistent. In addition, Table 8 shows that the instrumental variables are distinguishable owing to the Kleibergen–Paap rk Lagrange multiplier statistic of 72.95 (equivalent to a *p*-value of 0). As the value of the Cragg–Donald Wald F-statistic (83.32) is greater than the crucial value (16.38) for the Stock–Yogo weak ID test at the 10% confidence level, we can rule out the possibility of a weak instrumental variable.

### 4.4. Extensibility Study

In this study, we look at the impact performance of companies with different types of equity and different sector characteristics. The goal is to learn about the influence of green investment on corporate sustainability.

#### 4.4.1. Heterogeneity Analysis of the Nature of Equity

Table 9 presents the findings of the test for heterogeneity of the characteristics of equity. Green investment in the state-owned firm ($GI_{-1}$) model is 0.330, while that in the non-SOE ($GI_{-1}$) model is 2.532; both results are statistically significant at the 1% level. This result indicates that green investment by enterprises, regardless of whether they are state-owned or not, will improve their sustainability to some extent; however, some differences exist between them, as described in a number of studies [95], in which non-SOEs are found to be more effective in promoting sustainable development than SOEs in terms of investment in green innovation. In particular, a political connection exists between SOEs and the government, and relative to non-SOEs, SOEs tend to make green investments out of political and social responsibility and pay less attention to their sustainable development. Owing to the lack of innate support from the government, non-SOEs, on the one hand, can establish an informal relationship with the government through green investment to gain government attention and support. On the other hand, because private enterprises are self-financing, they consider the impact of multiple factors (e.g., internal investment costs and external financing constraints) when they make green investments. Accordingly, a company’s long-term objective when making environmentally responsible investments is to foster its continued growth and success.

#### 4.4.2. Heterogeneity Analysis of Industry

Despite their significant contributions to China’s economic growth, heavily polluting enterprises have led to weak endogenous growth and exacerbated ecological and environmental problems in society through their “three highs” (high energy consumption, high pollution, and high emissions) development approach. In this study, we do not group the sample size in the basic regression and moderating effect regression analyses, focusing on reducing sample specificity as much as possible in order to empirically analyze the impact of green investment behavior on the sustainable development of listed enterprises in Shanghai and Shenzhen A-shares. Considering that enterprises may affect the regression results due to the different industries they belong to, the total sample size of 14,779 enterprises participating in the regression is divided into highly polluting enterprises (5381) and non-highly polluting enterprises (9398) in the further research analysis in Section 4.4. Additionally, the division of the sample is based on the “Guidelines on Environmental Information Disclosure of Listed Companies” issued by the Chinese Ministry of Environmental Protection in 2010. The “Guidelines” classify mining, textiles, paper, paper products, petroleum, chemicals, chemical fiber, ferrous metals, and aluminum as highly polluting industries, and other industries as non-highly polluting industries, according to the industry distinction. On this basis, the group regressions were conducted with corporate lagged one-period green investment as the independent variable and corporate sustainability as the dependent variable, and the results in Table 10 were obtained.

The industry heterogeneity test results are presented in Table 10. According to the findings, the model of green investment of heavily polluting enterprises (${\mathrm{GI}}_{-1}$) with a coefficient of 1.974 and that of green investment of non-heavily polluting firms (${\mathrm{GI}}_{-1}$) with a coefficient of 1.312 are significant at the 1% level, indicating that the green investment of enterprises plays a significant role in promoting the sustainable development of enterprises. The coefficient of green investment is slightly higher for heavily polluting firms than for non-heavily polluting enterprises. As explained by Dang [96], enterprises in heavily polluting industries, whose environmental management effects are closely related to their own development, greatly need green investment. Meanwhile, heavily polluting enterprises are subject to strict government regulatory pressure and public supervision owing to their significant environmental hazards. Such characteristics facilitate the promotion of green investment, and the resulting improvement in sustainable development is increasingly significant.

## 5. Conclusions

### 5.1. Discussion

With the deterioration of the global ecological environment, the level of green investment and sustainability of companies, as important players contributing to socioeconomic development and environmental governance, are receiving increasing attention. Scholars such as Balcilar argue that green energy consumption and investment have a small but positive impact on economic growth, and their study concludes that the capacity utilization of green energy consumption and investment has not yet developed to a level that can mitigate the greenhouse effect and stimulate sustainable development in the long term to a viable level [16]. However, some scholars have argued that green investment has a key role in promoting high-quality economic development [17] and reducing carbon emissions [18]. In addition, many scholars also focus on the study of the relationship between environmental investment, financial performance, and environmental performance. For example, Zhang et al. argue that there is an inverse U-shape relationship between environmental performance and financial performance. [97]; Shabbir et al. argue that there is a significant positive relationship between intra-firm environmental investment and firm financial performance [98]. Li et al. argue that there is a positive correlation [99], etc.

The authors consider that in addition to improving financial performance and other indicators, the core focus of enterprise development should be on how to improve sustainability and thus improve market competitiveness and achieve green and virtuous development. The current moderating factors affecting the role of green investment in enterprise sustainable development are still unclear. In turn, the authors focus on the relationship between green investment in microenterprises and the achievement of corporate sustainable development goals.

We argue that analyzing the mechanisms influencing the role of corporate green investment on achieving corporate sustainable development goals from a stakeholder perspective is more conducive to accelerating the practice of green practices, increasing green investment, and achieving green development. The results of this study show that corporate green investment can significantly promote corporate sustainability; it also proves through empirical research that government environmental subsidies, investor attention, and executive overseas experience play a positive monitoring, motivating, and promoting role in corporate green investment for corporate sustainable development. This study has important theoretical and practical implications for how corporate green investment affects corporate sustainable development.

### 5.2. Conclusions

Using a sample of Chinese non-financial listed companies from 2010 to 2020, this study empirically investigates the contribution of green investment to corporate sustainable development and the moderating mechanisms of government environmental subsidies, investor attention, and executives’ overseas experience. The results reveal that green investment plays a significant role in promoting corporate sustainable development, thereby offering support to Hypothesis 1. Furthermore, government environmental protection subsidies, investor attention, and executives’ overseas experiences all have a positive moderating effect on green investment, promoting corporate sustainability. Finally, a study of firm ownership and industry heterogeneity shows that non-SOEs are more important than SOEs with regard to green investment for sustainable development. In addition, heavily polluting industries play a greater role in green investment for corporate sustainability than non-heavily polluting industries.

### 5.3. Implication of the Study

The findings of our study offer several theoretical and practical implications for advancing green investment and encouraging sustainable business growth.

(1) Theoretically, the results widen the linked theories of green development and environmental governance and deepen the linkage between green investment and sustainable development at the micro-level;

(2) In a practical sense, the results emphasize that businesses must first actively engage in green practices, uphold green management ideals, and enhance their green image so that green investment strategies can become an integral part of their long-term strategies and help them achieve sustainable development. Meanwhile, the government should increase environmental protection subsidies and formulate and adjust environmental regulation strategies in a reasonable and adaptable manner. On the one hand, such strategies can reduce the cost of green investment by enterprises, stabilize the normal operation of enterprises, ease financing constraints, and encourage enterprises to actively make green investments. On the other hand, the government’s publication of the green signal “compels” businesses to effectively conduct green investments. In addition, relevant government departments should introduce correspondingly favorable policies to further expand the scale of institutional investors. We should also use big data to grasp investor opinion in a timely manner, consider the impact of investor attention on the sustainable development of enterprises, direct investors toward clean energy, renewable resources, and other green projects, and pay attention to CSR. Finally, the role of globalization and the risk aversion of executives with overseas experience in environmental awareness should be stressed, and corporate leaders should be directed to consider sustainable development alongside short-term economic gains while making decisions;

(3) Furthermore, it facilitates the designation of appropriate policies by regulators and helps to develop some strategies and guidelines for emerging markets. First, the central government should strengthen environmental monitoring and implement differentiated incentives for green investments, as well as increase support for green technology innovation in manufacturing and private businesses, adapt taxation and other regulatory policies flexibly, and use emissions taxes to encourage high-polluting companies to invest in clean energy technologies. Second, managers and regulators should be aware of non-financial shareholder activism by investors, such as institutional investors’ voting on compensation and other monitoring activities, and fully utilize institutional investors’ role in keeping an eye on and directing businesses toward making green investments. Third, further optimize the strategy of introducing outstanding talents with overseas experience and attracting executives with overseas experience to participate in the core management of domestic enterprises through preferential welfare policies and compensation management, among other strategies, to improve the efficiency of green investment and the market competitiveness of domestic enterprises with their advanced environmental awareness and environmental vision accumulated overseas, thus driving the enterprises to achieve the goal of sustainable development.

### 5.4. Limitations and Future Directions

This study has some limitations with regard to the moderating effect found. Specifically, only three variables from typical stakeholder perspectives are selected to study the mechanism of green investment’s effect on corporate sustainability. On the basis of the previous literature, we learn that other micro- and macro-level factors influence corporate sustainable development; these factors include the degree of regional economic development, external financing restrictions, internal enterprise control, and the degree of enterprises’ green innovation. These factors are not fully covered in this study, but they can be explored more thoroughly and extensively in subsequent research.

In addition, the industry heterogeneity test reveals that heavily polluting industries are more effective at promoting sustainable development through green investment. In the next stage, more detailed research can be conducted on these industries to facilitate the provision of targeted and realistic guidance strategies for enterprises and to promote the overall green and sustainable development of society.

## Figures and Tables

**Figure 1 ijerph-20-01787-f001:**
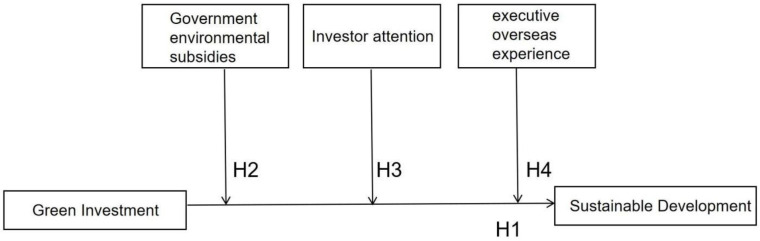
Study Model.

**Table 1 ijerph-20-01787-t001:** Variable names and definitions.

Variable	Symbol	Name	Definition
Explained variable	*SGR*	Sustainable Development	Net sales interest rate × total asset turnover × income retention rate × equity multiplier/(1—net sales interest rate × total asset turnover × income retention rate × equity multiplier)
Explanatory variable	*GI*	Green Investment	Total environmental investment/total assets at year-end
Moderating variable	*ENVSUB*	Government Environmental Subsidies	Total environmental subsidies/current year operating revenue
	*ATTEN*	Investor Attention	Total number of Google web searches for stock symbols/million
	*OVERSEA*	Executives’ Overseas Experience	Number of executives with overseas experience/total number of executives
Controlled variable	*SIZE*	Company Size	Natural logarithm of total assets for the year
	*LEV*	Asset–Liability Ratio	Total liabilities/total assets
	*ROA*	Profitability	Net income/average balance of total assets
	*ATO*	Net Asset Turnover Ratio	Sales revenue/average of total net assets at the beginning and end of the period
	*CFLOW*	Cash Flow Ratio	Net cash flow from operating activities/total assets
	*GROWTH*	Growth	Main business income of this period/main business income of period−1
	*TOBINQ*	Tobin Q	(Outstanding market value + non-marketable par value/total assets − net intangible assets − net goodwill)
	*SOE*	Nature of Equity	State-controlled = 1; otherwise = 0
	*AGE*	Enterprise Age	Ln (enterprise observation year − registration year + 1)
	*INDUSTRY*	Industry	Industry dummy variable, belonging to the corresponding year = 1, otherwise = 0
	*YEAR*	Year	Time dummy variable, belonging to the corresponding year = 1, otherwise = 0

**Table 2 ijerph-20-01787-t002:** Results of the annual sample size distribution.

Year	2010	2011	2012	2013	2014	2015	2016	2017	2018	2019	2020
Sample	N = 0	N = 999	N = 1157	N = 1294	N = 1337	N = 1236	N = 1333	N = 1586	N = 1764	N = 2016	N = 2057

**Table 3 ijerph-20-01787-t003:** Results of the descriptive statistics.

Variable	N	Mean	SD	Min	p50	Max	Skewness	Kurtosis
*SGR*	14,779	0.031	0.078	−0.057	0.008	0.633	5.196	35.499
*GI*	14,779	0.045	0.005	0.034	0.045	0.057	0.033	2.540
*ENVSUB*	14,779	0.046	0.016	0.018	0.043	0.119	1.614	7.182
*ATTEN*	14,779	1.091	0.558	0.500	0.910	2.480	0.857	2.647
*OVERSEA*	14,779	0.071	0.082	0.000	0.053	0.385	1.501	5.309
*SIZE*	14,779	22.424	1.328	20.029	22.216	26.272	0.747	3.287
*LEV*	14,779	0.422	0.197	0.055	0.417	0.857	0.149	2.190
*ROA*	14,779	0.054	0.044	0.002	0.043	0.221	1.399	5.105
*ATO*	14,779	1.313	1.089	0.148	1.006	6.888	2.454	10.775
*CFLOW*	14,779	0.055	0.066	−0.138	0.054	0.240	0.020	3.711
*GROWTH*	14,779	0.174	0.365	−0.427	0.111	2.744	3.610	23.160
*TOBINQ*	14,779	1.948	1.195	0.860	1.554	7.915	2.484	10.605
*SOE*	14,779	0.400	0.490	0.000	0.000	1.000	0.410	1.169
*AGE*	14,779	2.878	0.335	1.609	2.944	3.466	−0.915	4.227

**Table 4 ijerph-20-01787-t004:** Results of the correlation analysis.

	*SGR*	*GI*	*ENVSUB*	*ATTEN*	*OVERSEA*	*SIZE*	*LEV*	*ROA*	*ATO*	*CFLOW*	*GROWTH*	*TOBINQ*	*SOE*	*AGE*
*SGR*	1													
*GI*	0.174 ***	1												
*ENVSUB*	0.310 ***	0.204 ***	1											
*ATTEN*	0.145 ***	0.175 ***	0.191 ***	1										
*OVERSEA*	0.207 ***	0.196 ***	0.210 ***	0.260 ***	1									
*SIZE*	0.117 ***	0.048 ***	0.078 ***	0.199 ***	0.110 ***	1								
*LEV*	0.110 ***	0.049 ***	0.142 ***	0.062 ***	0.003	0.567 ***	1							
*ROA*	0.234 ***	0.058 ***	0.010	0.027 ***	0.140 ***	−0.098 ***	−0.392 ***	1						
*ATO*	−0.052 ***	0.012	−0.043 ***	−0.009	−0.036 ***	0.251 ***	0.511 ***	−0.055 ***	1					
*CFLOW*	−0.051 ***	−0.032 ***	−0.126 ***	−0.024 ***	0.026 ***	0.024 ***	−0.186 ***	0.471 ***	0.021 **	1				
*GROWTH*	0.328 ***	0.101 ***	0.224 ***	0.047 ***	0.089 ***	0.017 **	0.067 ***	0.170 ***	0.101 ***	−0.014 *	1			
*TOBINQ*	0.038 ***	0.004	−0.019 **	0.036 ***	0.079 ***	−0.382 ***	−0.359 ***	0.379 ***	−0.123 ***	0.173 ***	0.037 ***	1		
*SOE*	−0.037 ***	0.005	0.016 **	0.046 ***	−0.181 ***	0.388 ***	0.316 ***	−0.182 ***	0.180 ***	−0.043 ***	−0.074 ***	−0.177 ***	1	
*AGE*	0.004	−0.022 ***	0.022 ***	−0.014 *	−0.051 ***	0.167 ***	0.142 ***	−0.054 ***	0.026 ***	0.045 ***	−0.056 ***	−0.031 ***	0.178 ***	1

Note: ***, **, and * denote significance at the 1%, 5%, and 10% levels, respectively.

**Table 5 ijerph-20-01787-t005:** Results of the regression analysis.

	(1)	(2)	(3)	(4)
Variables	*SGR*	*SGR*	*SGR*	*SGR*
*GI_−1_*	1.563 ***	1.142 ***	1.186 ***	0.984 ***
	(8.10)	(7.36)	(7.09)	(6.28)
*SIZE*	0.016 ***	0.016 ***	0.014 ***	0.014 ***
	(5.13)	(5.42)	(4.67)	(4.64)
*LEV*	0.131 ***	0.113 ***	0.130 ***	0.117 ***
	(9.59)	(8.74)	(9.60)	(8.80)
*ROA*	0.727 ***	0.652 ***	0.685 ***	0.619 ***
	(15.71)	(15.10)	(15.22)	(13.99)
*ATO*	−0.018 ***	−0.016 ***	−0.018 ***	−0.017 ***
	(−9.07)	(−8.36)	(−8.56)	(−8.67)
*GROWTH*	0.044 ***	0.036 ***	0.043 ***	0.042 ***
	(8.60)	(7.23)	(8.53)	(8.62)
*TOBIN*	−0.002 **	−0.002 **	−0.003 ***	−0.002 **
	(−2.48)	(−2.26)	(−3.02)	(−2.12)
*AGE*	0.003	0.001	0.019 *	0.007
	(0.23)	(0.11)	(1.65)	(0.65)
*CFLOW*	−0.121 ***	−0.097 ***	−0.111 ***	−0.105 ***
	(−7.10)	(−6.26)	(−6.75)	(−6.76)
*SOE*	−0.011 **	−0.009 *	−0.007	−0.003
	(−2.31)	(−1.80)	(−1.39)	(−0.63)
*ENVSUB*		0.713 ***		
		(8.33)		
*GI_−1_* *∗ ENVSUB*		0.013 ***		
		(8.59)		
*ATTEN*			0.024 ***	
			(6.03)	
*GI_−1_* *∗ ATTEN*			0.008 ***	
			(6.80)	
*OVERSEA*				0.189 ***
				(8.55)
*GI_−1_* *∗ OVERSEA*				0.012 ***
				(8.35)
_cons	−0.447 ***	−0.461 ***	−0.469 ***	−0.397 ***
	(−6.69)	(−7.12)	(−6.86)	(−6.26)
Firm	Yes	Yes	Yes	Yes
Industry	Yes	Yes	Yes	Yes
Year	Yes	Yes	Yes	Yes
N	14,779	14,779	14,779	14,779
R^2^	0.198	0.262	0.223	0.251
adj. R^2^	0.197	0.261	0.222	0.250
F	32.778	38.123	32.861	37.807

Notes: *t*-statistics in parentheses; *** *p <* 0.01, ** *p* < 0.05, and * *p* < 0.1.

**Table 6 ijerph-20-01787-t006:** Robustness test: regression results of the adjusted sample interval.

	(1)	(2)	(3)	(4)
Variables	*SGR*	*SGR*	*SGR*	*SGR*
*GI_−1_*	1.738 ***	1.290 ***	1.283 ***	1.107 ***
	(8.08)	(7.31)	(6.81)	(6.19)
*SIZE*	0.017 ***	0.017 ***	0.016 ***	0.015 ***
	(5.05)	(5.08)	(4.52)	(4.63)
*LEV*	0.137 ***	0.118 ***	0.134 ***	0.121 ***
	(9.03)	(8.24)	(9.04)	(8.24)
*ROA*	0.730 ***	0.647 ***	0.679 ***	0.612 ***
	(14.39)	(13.77)	(13.86)	(12.70)
*ATO*	−0.019 ***	−0.017 ***	−0.018 ***	−0.017 ***
	(−8.11)	(−7.45)	(−7.71)	(−7.87)
*GROWTH*	0.044 ***	0.035 ***	0.043 ***	0.042 ***
	(8.08)	(6.67)	(8.07)	(8.09)
*TOBIN*	−0.002	−0.002	−0.002 **	−0.001
	(−1.57)	(−1.45)	(−2.19)	(−1.35)
*AGE*	−0.003	−0.004	0.012	0.003
	(−0.21)	(−0.37)	(1.51)	(0.29)
*CFLOW*	−0.117 ***	−0.093 ***	−0.105 ***	−0.099 ***
	(−6.42)	(−5.44)	(−5.91)	(−5.92)
*SOE*	−0.015 **	−0.015 **	−0.011 *	−0.005
	(−2.55)	(−2.34)	(−1.74)	(−0.86)
*ENVSUB*		0.772 ***		
		(8.26)		
*GI*_−1_ ∗ *ENVSUB*		0.013 ***		
		(8.23)		
*ATTEN*			0.028 ***	
			(6.61)	
*GI*_−1_ ∗ *ATTEN*			0.008 ***	
			(6.34)	
*OVERSEA*				0.196 ***
				(8.34)
*GI*_−1_ ∗ *OVERSEA*				0.012 ***
				(8.46)
_cons	−0.470 ***	−0.472 ***	−0.503 ***	−0.420 ***
	(−6.25)	(−6.44)	(−6.46)	(−5.95)
Firm	Yes	Yes	Yes	Yes
Industry	Yes	Yes	Yes	Yes
Year	Yes	Yes	Yes	Yes
N	12,722	12,722	12,722	12,722
R^2^	0.201	0.269	0.230	0.259
adj. R^2^	0.199	0.268	0.228	0.258
F	31.477	37.741	31.851	36.852

Notes: t-statistics in parentheses; *** *p <* 0.01, ** *p* < 0.05, and * *p* < 0.1.

**Table 7 ijerph-20-01787-t007:** Robustness test: regression results of the adjusted range of independent variables.

	(1)	(2)	(3)	(4)
Variables	*SGR*	*SGR*	*SGR*	*SGR*
*GIINV_−1_*	1.833 ***	1.201 ***	1.184 ***	0.847 ***
	(6.52)	(4.95)	(4.58)	(3.49)
*SIZE*	0.016 ***	0.017 ***	0.015 ***	0.014 ***
	(5.23)	(5.58)	(4.75)	(4.59)
*LEV*	0.132 ***	0.114 ***	0.131 ***	0.121 ***
	(9.56)	(8.67)	(9.57)	(8.91)
*ROA*	0.731 ***	0.666 ***	0.692 ***	0.632 ***
	(15.61)	(15.26)	(15.11)	(14.08)
*ATO*	−0.018 ***	−0.016 ***	−0.018 ***	−0.017 ***
	(−9.14)	(−8.20)	(−8.47)	(−8.75)
*GROWTH*	0.045 ***	0.037 ***	0.044 ***	0.043 ***
	(8.69)	(7.42)	(8.64)	(8.72)
*TOBIN*	−0.002 **	−0.002 **	−0.003 ***	−0.002 **
	(−2.35)	(−2.09)	(−3.00)	(−2.27)
*AGE*	0.001	−0.002	0.020 *	0.009
	(0.09)	(−0.18)	(1.73)	(0.77)
*CFLOW*	−0.122 ***	−0.102 ***	−0.115 ***	−0.107 ***
	(−7.08)	(−6.47)	(−6.87)	(−6.76)
*SOE*	−0.011 **	−0.010 *	−0.006	−0.004
	(−2.14)	(−1.82)	(−1.18)	(−0.69)
*ENVSUB*		0.791 ***		
		(8.50)		
*GIINV_−1_* *∗ENVSUB*		0.011 ***		
		(7.67)		
*ATTEN*			0.027 ***	
			(6.27)	
*GIINV_−1_* *∗ ATTEN*			0.007 ***	
			(6.75)	
*OVERSEA*				0.215 ***
				(8.34)
*GIINV_−1_* *∗ OVERSEA*				0.010 ***
				(8.60)
_cons	−0.412 ***	−0.436 ***	−0.450 ***	−0.374 ***
	(−6.19)	(−6.75)	(−6.50)	(−5.82)
Firm	Yes	Yes	Yes	Yes
Industry	Yes	Yes	Yes	Yes
Year	Yes	Yes	Yes	Yes
N	14,779	14,779	14,779	14,779
R^2^	0.191	0.244	0.214	0.239
adj. R^2^	0.190	0.243	0.213	0.238
F	32.737	36.407	32.177	35.488

Notes: *t*-statistics in parentheses; *** *p <* 0.01, ** *p* < 0.05, and * *p* < 0.1.

**Table 8 ijerph-20-01787-t008:** Heterogeneity test: two-stage least squares regression results.

	First Stage	Second Stage
Variables	*GI_−1_*	*SGR*
*GI_−1_*		4.426 ***
		2.72
*GImean_−1_*	1.056 ***	
	9.14	
*SIZE*	0.001 ***	0.014 ***
	4.23	4.14
*LEV*	0.003 ***	0.123 ***
	3.95	8.73
*ROA*	0.014 ***	0.686 ***
	6.51	13.21
*ATO*	−0.000 ***	−0.017 ***
	(−3.18)	(−8.44)
*GROWTH*	0.001 ***	0.041 ***
	5.86	7.73
*TOBIN*	0.000	−0.003 ***
	0.81	(−2.60)
*AGE*	−0.002 ***	0.008
	(−2.84)	0.64
*CFLOW*	−0.003 ***	−0.111 ***
	(−3.62)	(−6.20)
*SOE*	0.000	−0.013 **
	1.11	(−2.50)
Firm	YES	YES
Industry	YES	YES
Year	YES	YES
Observations	14,779	14,779
R-squared		0.158
Number of id	2513	2513
F	83.61	32.72
Kleibergen–Paap rk LM statistic	72.95(Chi-sq(1) *p*-value = 0.0000)	
Cragg–Donald F statistic	83.32	
Kleibergen–Paap rk Wald F statistic	83.61	
10% maximal instrumental variable size	16.38	

Notes: *t*-statistics in parentheses; *** *p <* 0.01, ** *p* < 0.05.

**Table 9 ijerph-20-01787-t009:** Heterogeneity test: nature of equity.

	State-Owned Enterprise	Non-State-Owned Enterprise
	*SGR*	*SGR*
*GI_−1_*	0.330 *	2.532 ***
	(1.68)	(8.44)
*SIZE*	0.011 **	0.020 ***
	(2.20)	(4.74)
*LEV*	0.136 ***	0.135 ***
	(5.51)	(7.87)
*ROA*	0.695 ***	0.731 ***
	(9.80)	(12.13)
*ATO*	−0.015 ***	−0.025 ***
	(−6.22)	(−6.35)
*GROWTH*	0.051 ***	0.040 ***
	(6.23)	(6.05)
*TOBIN*	−0.002	−0.003 **
	(−1.38)	(−2.24)
*AGE*	−0.019	0.016
	(−1.11)	(0.98)
*CFLOW*	−0.107 ***	−0.119 ***
	(−4.15)	(−5.27)
_cons	−0.248 **	−0.603 ***
	(−2.26)	(−6.60)
Firm	Yes	Yes
Industry	Yes	Yes
Year	Yes	Yes
N	5904	8875
R^2^	0.168	0.228
adj. R^2^	0.166	0.226
F	12.500	24.264

Notes: *t*-statistics in parentheses; *** *p <* 0.01, ** *p* < 0.05, and * *p* < 0.1.

**Table 10 ijerph-20-01787-t010:** Heterogeneity test: industry classification.

	Heavily Polluting Industry	Non-Heavily Polluting Industry
	*SGR*	*SGR*
*GI_−1_*	1.974 ***	1.312 ***
	(5.92)	(5.58)
*SIZE*	0.020 ***	0.013 ***
	(4.30)	(3.33)
*LEV*	0.090 ***	0.164 ***
	(4.88)	(8.39)
*ROA*	0.638 ***	0.789 ***
	(8.61)	(13.48)
*ATO*	−0.013 ***	−0.022 ***
	(−5.26)	(−7.51)
*GROWTH*	0.049 ***	0.041 ***
	(5.24)	(6.89)
*TOBIN*	−0.004 *	−0.002
	(−1.75)	(−1.59)
*AGE*	0.030	−0.012
	(1.55)	(−0.87)
*CFLOW*	−0.095 ***	−0.132 ***
	(−4.93)	(−5.74)
*SOE*	−0.001	−0.016 **
	(−0.06)	(−2.56)
_cons	−0.630 ***	−0.347 ***
	(−6.07)	(−4.06)
Firm	Yes	Yes
Industry	Yes	Yes
Year	Yes	Yes
N	5381	9398
R^2^	0.230	0.188
adj. R^2^	0.227	0.186
F	13.425	21.891

Notes: *t*-statistics in parentheses; *** *p <* 0.01, ** *p* < 0.05, and **p* < 0.1.

## Data Availability

The raw data supporting the conclusions of this article will be made available by the authors without undue reservation.

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
