# Peer review of "Can Enterprises in China Achieve Sustainable Development through Green Investment?"

_ijerph, 2023, doi:10.3390/ijerph20031787_

Round 1
Reviewer 1 Report
Well, the overall draft of the paper is a reasonable effort by the authors, but I have some major suggestions:
1. The introduction is ambiguous, so it should be re-written well, i.e., it is suggested to move the theories to the literature review section and focus more on the significance of the study with strong arguments and main contributions of the study toward literature in a concise manner.
2. On Page# 7, line #293, the authors need to mention how much winsorized the data.
3. Table#2's descriptive statistic includes skewness and kurtosis values.
4. In the conclusion section, the policy implications for the policy makers or regulators and the future direction for the scholars should be addressed. Further, also provide how the findings of the current study help to make some strategies and guideline for emerging markets.
5. The authors should create a link between the results obtained in the study and the results presented in the Literature review.
6. Finally, it is suggested that cite more relevant publications from 2022 to update the literature review section. You may take help from these recent relevant studies**
**a). Ren, S., Hao, Y., & Wu, H. (2022). How does green investment affect environmental pollution? Evidence from China. Environmental and Resource Economics, 81(1), 25-51. https://doi.org/10.1007/s10640-021-00615-4
b). Zheng, J., Khurram, M. U., & Chen, L. (2022). Can Green Innovation Affect ESG Ratings and Financial Performance? Evidence from Chinese GEM Listed Companies. Sustainability, 14(14), 8677. https://doi.org/10.3390/su14148677
c). Li, G., Zhang, R., Feng, S., & Wang, Y. (2022). Digital finance and sustainable development: Evidence from environmental inequality in China. Business Strategy and the Environment.https://doi.org/10.1002/bse.3105
d). Balcilar, M., Usman, O., & Ike, G. N. (2022). Investing green for sustainable development without ditching economic growth. Sustainable Development. https://doi.org/10.1002/sd.2415
e). Irfan, M., Razzaq, A., Sharif, A., & Yang, X. (2022). Influence mechanism between green finance and green innovation: Exploring regional policy intervention effects in China. Technological Forecasting and Social Change, 182, 121882.https://doi.org/10.1016/j.techfore.2022.121882
Author Response
Well, the overall draft of the paper is a reasonable effort by the authors, but I have some major suggestions:
- The introduction is ambiguous, so it should be re-written well, i.e., it is suggested to move the theories to the literature review section and focus more on the significance of the study with strong arguments and main contributions of the study toward literature in a concise manner.
Reply: According to the reviewer’s comments, we have moved the theories to the literature review section and added the contents in the introduction as follows: (p2-p3)
The Chinese government has used administrative regulations, economic instruments, and market mechanisms with some success to limit pollutant emissions [13], but if serious problems such as resource scarcity, environmental pollution, and ecological damage are not fundamentally addressed, China will struggle to achieve its green and sustainable development goals [14]. Environmental equity, according to Li and other scholars, is essential for achieving resource use efficiency and sustainable development, while local government environmental governance capacity and corporate green innovation technologies can effectively mitigate environmental inequality [15]. According to Ren et al., green investment can reduce environmental pollution by enhancing energy conservation and emission reduction capacities, expanding technological innovation capacities, and modernizing industrial structures [13]. Green energy consumption and investment have a small but positive impact on economic growth, according to Balcilar and other macroeconomists [16]. However, some academics are optimistic about the impact of green investment on economic development. The literature has evaluated the influence of green investment on high-quality economic development and carbon emission reduction, concluding that green investment has become crucial in promoting high-quality economic development [17] and reducing carbon emissions [18].
Based on the analysis of the literature and the current research context, we focus not only on the impact of green investment on energy savings, emission reduction, and environmental management, but also on how micro enterprises can achieve their own sustainable development goals through green investment and then promote the realization of a virtuous cycle of corporate environmental spending and sustainable development. To fill this research gap, this study constructs a study of Chinese listed companies in the Shanghai and Shenzhen stock markets from 2010 to 2020. The impact of the green investment behavior of Chinese firms on improving corporate sustainability is empirically examined, and the results show that green investment by Chinese firms can contribute to enhancing corporate sustainability. This study further explores the mechanism of the interaction between the two using the moderating effect. The empirical study finds that government environmental subsidies, investor attention, and executive overseas experience can positively moderate the contribution of green investment to corporate sustainability. It provides an important theoretical reference for enterprises to increase green investment and achieve sustainable development goals. It also provides strong theoretical support and policy guidance for regulators, investors, and executives on how to help enterprises achieve sound economic and environmental sustainability.
- On Page# 7, line #293, the authors need to mention how much winsorized the data.
Reply: According to the reviewer’s comments, we have revised the contents as follows: (p8)
Fifth, to reduce the effect of outliers, the variables were winsorized at the upper and lower 1% levels (excluding the dummy variables and corporate age);
- Table#2's descriptive statistic includes skewness and kurtosis values.
Reply: According to the reviewer’s comments, we have added skewness and kurtosis values in the Table. (p12)
Table 3. Results of the descriptive statistics.
|
Variable |
N |
Mean |
SD |
Min |
p50 |
Max |
Skewness |
Kurtosis |
|
SGR |
14779 |
0.031 |
0.078 |
-0.057 |
0.008 |
0.633 |
5.196 |
35.499 |
|
GI |
14779 |
0.045 |
0.005 |
0.034 |
0.045 |
0.057 |
0.033 |
2.540 |
|
ENVSUB |
14779 |
0.046 |
0.016 |
0.018 |
0.043 |
0.119 |
1.614 |
7.182 |
|
ATTEN |
14779 |
1.091 |
0.558 |
0.500 |
0.910 |
2.480 |
0.857 |
2.647 |
|
OVERSEA |
14779 |
0.071 |
0.082 |
0.000 |
0.053 |
0.385 |
1.501 |
5.309 |
|
SIZE |
14779 |
22.424 |
1.328 |
20.029 |
22.216 |
26.272 |
0.747 |
3.287 |
|
LEV |
14779 |
0.422 |
0.197 |
0.055 |
0.417 |
0.857 |
0.149 |
2.190 |
|
ROA |
14779 |
0.054 |
0.044 |
0.002 |
0.043 |
0.221 |
1.399 |
5.105 |
|
ATO |
14779 |
1.313 |
1.089 |
0.148 |
1.006 |
6.888 |
2.454 |
10.775 |
|
CFLOW |
14779 |
0.055 |
0.066 |
-0.138 |
0.054 |
0.240 |
0.020 |
3.711 |
|
GROWTH |
14779 |
0.174 |
0.365 |
-0.427 |
0.111 |
2.744 |
3.610 |
23.160 |
|
TOBINQ |
14779 |
1.948 |
1.195 |
0.860 |
1.554 |
7.915 |
2.484 |
10.605 |
|
SOE |
14779 |
0.400 |
0.490 |
0.000 |
0.000 |
1.000 |
0.410 |
1.169 |
|
AGE |
14779 |
2.878 |
0.335 |
1.609 |
2.944 |
3.466 |
-0.915 |
4.227 |
- In the conclusion section, the policy implications for the policy makers or regulators and the future direction for the scholars should be addressed. Further, also provide how the findings of the current study help to make some strategies and guideline for emerging markets.
Reply: According to the reviewer’s comments, we have added the contents as follows: (p22)
With the deterioration of the global ecological environment, the level of green investment and sustainability of companies, as important players contributing to socioeconomic development and environmental governance, are receiving increasing attention. Scholars such as Balcilar argue that green energy consumption and investment have a small but positive impact on economic growth, and their study concludes that the capacity utilization of green energy consumption and investment has not yet developed to a level that can mitigate the greenhouse effect and stimulate sustainable development in the long term to a viable level [16]. However, some scholars have argued that green investment has a key role in promoting high-quality economic development [17] and reducing carbon emissions [18]. In addition, many scholars also focus on the study of the relationship between environmental investment and financial performance and environmental performance. For example, Jin et al. argue that there is a u-shaped relationship between environmental investment and financial performance [97]; Shabbir et al. argue that there is a significant positive relationship between intra-firm environmental investment and firm financial performance [98]. Li et al. argue that there is a positive correlation [99], etc.
The authors consider that in addition to improving financial performance and other indicators, the core focus of enterprise development should be on how to improve its sustainability and thus improve market competitiveness and achieve green and virtuous development. The current moderating factors affecting the role of green investment in enterprise sustainable development are still unclear. In turn, the authors focus on the relationship between green investment in micro enterprises and the achievement of corporate sustainable development goals.
- The authors should create a link between the results obtained in the study and the results presented in the Literature review.
Reply: According to the reviewer’s comments, we have added the contents as follows:
Studies based on neoclassical economic theory argue that corporate environmental protection expenditures crowd out productive capital, leading to a decrease in productivity and profit [25]. Green investment by businesses is a "passive activity" that complies with governments' environmental rules [26]. This characterization is due to the high barrier to entry for businesses looking to invest in environmental safeguards, such as the purchase of environmental protection equipment and funding of green technology research and development (R&D), and the low probability of a positive return [27]. According to studies based on the Porter hypothesis, environmental investments raise expenses in the short term but have the potential to spur technical innovation and generate "innovation compensation" over the long term, thereby improving productivity and performance [28]. Meanwhile, green investment has also been regarded as an "active behavior" of enterprises. On the one hand, businesses benefit from increasing the rate at which they use their resources, attracting more customers and other sources of funding, building a positive reputation among their target audiences, and achieving sustainable development [29]. Taylor et al. concluded that upholding environmental responsibility may result in a short-term loss of economic benefits but guarantees long-term sustainable development [30]. On the other hand, businesses can boost customer loyalty and reputation by engaging in green practices, thus reaping economic benefits from an indirect approach [31]. (p4)
According to Brown et al., government taxes encourage businesses that produce pollution to invest in adopting and using cleaner production methods. They contend that taxing pollutant emissions encourages businesses to invest more in green technologies to switch to cleaner production. Green investments, they claim, have higher marginal returns for polluting businesses [40] and are more likely to encourage long-term corporate growth. (p4)
Empirical evidence from Sun et al. supports the notion that green investments and clean energy both play significant roles in reducing environmental pollution and fostering environmentally sustainable development [43]. Xin et al. found that CSR can promote high-quality corporate development through improved green innovation, environmental investment, and corporate governance [44]. (p4)
Liu and other scholars concluded from empirical analysis that environmental taxes have a catalytic effect on firms to increase their environmental investments and that environmental taxes promote firm performance by increasing environmental investments [48]. (p5)
Using empirical analysis, Liu et al. came to the conclusion that environmental taxes improve firms' performance by spurring increased environmental investment and that environmental taxes have a catalytic effect on this increase in investment [48]. Government environmental spending, according to Hong et al., can lead to businesses adopting a free-rider mentality and avoiding environmental responsibility, which has a detrimental effect on how effective that spending is. Corporate green investment is negatively impacted by government environmental spending, or "crowding out" [56]. However, Ding et al. came to the conclusion that government subsidies have a favorable impact on corporate environmental investment [57] and that the association between government subsidies and corporate environmental investment can help decision-makers develop policies and allocate limited resources. (p6)
Deng et al. discovered that the occurrence of environmental events may strengthen firms' environmental investments and that extremely high investor attention leads to higher returns for green firms [59]. This, in turn, may lead firms to increase their green investments because of the attention that investors pay to green companies. (p6)
Othar Kordsachia and colleagues argued that the propensity for sustainability held by institutional investors is also a driving force behind firms' green investments. They argue that companies with more sustainable institutional investor ownership have higher carbon risk awareness and that sustainable institutional ownership is positively related to firms' environmental performance [64]. In addition, they claim that firms with more sustainable institutional investor ownership have higher environmental performance. According to the findings of Pan et al.'s research, public concern for the environment has a positive impact on the effectiveness of green investments made by businesses [65]. (p7)
Cao et al. argued that executives with overseas experience have substantial beneficial effects on the behavioral decisions of companies' green investment and environmental social responsibility [67], and how to make use of executives with overseas backgrounds to improve the efficiency of enterprises' green investment becomes an important issue. Cui et al. discovered that, when comparing the roles of executives with overseas backgrounds in high and low carbon firms, executives with overseas experience play a larger role in green innovation in high carbon firms and are more conducive to meeting carbon reduction goals [68] in response to social environment needs [69]. (p7)
Zhang et al. concluded that the overseas backgrounds of CEO-led companies provide higher quality sustainability reports [71], suggesting that executives with overseas backgrounds have higher ethics, integrity, and focus on sustainability. (p7)
Chen et al. concluded that the overseas background of executives has a significant positive impact on the green technology innovation of enterprises, and even its effect on the green innovation of enterprises is more obvious in high pollution cities [79]. (p7)
With the deterioration of the global ecological environment, the level of green investment and sustainability of companies, as important players contributing to socioeconomic development and environmental governance, are receiving increasing attention. scholars such as Balcilar argue that green energy consumption and investment have a small but positive impact on economic growth, and their study concludes that the capacity utilization of green energy consumption and investment has not yet developed to a level that can mitigate greenhouse effect and stimulate sustainable development in the long term to a viable level [87]. However, some scholars have argued that green investment has a key role in promoting high-quality economic development [17] and reducing carbon emissions [18]. In addition, many scholars also focus on the study of the relationship between environmental investment and financial performance and environmental performance, for example, Jin et al. argue that there is a u-shaped relationship between environmental investment and financial performance [88]; Shabbir et al. argue that there is a significant positive relationship between intra-firm environmental investment and firm financial performance [89]. Li et al. argue that there is a positive correlation [90], etc. (p22)
- Finally, it is suggested that cite more relevant publications from 2022 to update the literature review section. You may take help from these recent relevant studies**
**a). Ren, S., Hao, Y., & Wu, H. (2022). How does green investment affect environmental pollution? Evidence from China. Environmental and Resource Economics, 81(1), 25-51. https://doi.org/10.1007/s10640-021-00615-4
b). Zheng, J., Khurram, M. U., & Chen, L. (2022). Can Green Innovation Affect ESG Ratings and Financial Performance? Evidence from Chinese GEM Listed Companies. Sustainability, 14(14), 8677. https://doi.org/10.3390/su14148677
c). Li, G., Zhang, R., Feng, S., & Wang, Y. (2022). Digital finance and sustainable development: Evidence from environmental inequality in China. Business Strategy and the Environment.https://doi.org/10.1002/bse.3105
d). Balcilar, M., Usman, O., & Ike, G. N. (2022). Investing green for sustainable development without ditching economic growth. Sustainable Development. https://doi.org/10.1002/sd.2415
e). Irfan, M., Razzaq, A., Sharif, A., & Yang, X. (2022). Influence mechanism between green finance and green innovation: Exploring regional policy intervention effects in China. Technological Forecasting and Social Change, 182, 121882.https://doi.org/10.1016/j.techfore.2022.121882
Reply: According to the reviewer’s comments, we have added the references as follows:
- Ren, S.; Hao, Y.; Wu, H. How Does Green Investment Affect Environmental Pollution? Evidence from China. Environmental and Resource Economics, 2022, 81(1), 25-51. [CrossRef]
- Li, G.; Zhang, R.; Feng, S.; Wang, Y. Digital Finance and Sustainable Development: Evidence from Environmental Inequality in China. Strategy Environ. 2022, 31 (7), 3574-3594. [CrossRef]
- Balcilar, M.; Usman, O.; Ike, G. N. Investing Green for Sustainable Development without Ditching Economic G Sustainable Development, 2022. [CrossRef]
- Zheng, J.; Khurram, M. U.; Chen, L. Can Green Innovation Affect ESG Ratings and Financial Performance? Evidence from Chinese GEM Listed Companies. Sustainability 2022, 14 (14), 8677. [CrossRef]

Reviewer 2 Report
Overall, I think this is a solid contribution to the literature. The topic is important from the perspective of sustainable the companies.
· The study is extensively well described (introduction, method, results).
· I miss a discussion part.
· Great that you link to the Sustainable Development Goals. Can you mention briefly some examples? SDG 12 for example.
· Do public agencies control companies regarding the rules to be environmentally friendly? How?
1. Introduction
· Raw 161, explain green technology spillovers. What kind of effects do companies have?
2. Results
· Is it possible to build extreme groups – calculate to upper and lower percentile of 20%? Your sample size is huge.
· Regarding on the sample size, is it possible to split into different group? Are there different between males and females? Are there differences between the companies exist? à e.g. mining, textile, petroleum, chemical etc. are there any differences between them?
· How did Corona affect the results? Past – present – are there any differences?
· Row 567-69 how do you choose both groups? Sample size of both groups? Even so Table 9 what is the variable to split into polluting/non-heavily polluted industry?
· Do you include a control group or is it possible to supply?
· 10 years – I miss a overview like a table of year and sample size
|
2010 |
2011 |
2012 |
… |
2020 |
|
N=XX |
|
|
|
|
à separating the industry, highlighting effects like the pandemic. Over the years, are the participants the same?
3. Discussion
· Overall, I miss some links to other studies relating to your findings.
Author Response
Overall, I think this is a solid contribution to the literature. The topic is important from the perspective of sustainable the companies.
- The study is extensively well described (introduction, method, results).
- I miss a discussion part.
Reply: According to the reviewer’s comments, we have added the discussion part. (p22-p23)
5.1 Discussion
With the deterioration of the global ecological environment, the level of green investment and sustainability of companies, as important players contributing to socioeconomic development and environmental governance, are receiving increasing attention. Scholars such as Balcilar argue that green energy consumption and investment have a small but positive impact on economic growth, and their study concludes that the capacity utilization of green energy consumption and investment has not yet developed to a level that can mitigate the greenhouse effect and stimulate sustainable development in the long term to a viable level [16]. However, some scholars have argued that green investment has a key role in promoting high-quality economic development [17] and reducing carbon emissions [18]. In addition, many scholars also focus on the study of the relationship between environmental investment and financial performance and environmental performance. For example, Jin et al. argue that there is a u-shaped relationship between environmental investment and financial performance [97]; Shabbir et al. argue that there is a significant positive relationship between intra-firm environmental investment and firm financial performance [98]. Li et al. argue that there is a positive correlation [99], etc.
The authors consider that in addition to improving financial performance and other indicators, the core focus of enterprise development should be on how to improve its sustainability and thus improve market competitiveness and achieve green and virtuous development. The current moderating factors affecting the role of green investment in enterprise sustainable development are still unclear. In turn, the authors focus on the relationship between green investment in micro enterprises and the achievement of corporate sustainable development goals.
We argue that analyzing the mechanisms influencing the role of corporate green investment on achieving corporate sustainable development goals from a stakeholder perspective is more conducive to accelerating the practice of green practices, increasing green investment, and achieving green development. The results of this study show that corporate green investment can significantly promote corporate sustainability; it also proves through empirical research that government environmental subsidies, investor attention, and executive overseas experience play a positive monitoring, motivating, and promoting role in corporate green investment for corporate sustainable development. This study has important theoretical and practical implications for how corporate green investment affects corporate sustainable development.
- Great that you link to the Sustainable Development Goals. Can you mention briefly some examples? SDG 12 for example.
Reply: According to the reviewer’s comments, we have added the contents as follows: (p1-p2)
The United Nations convened the Sustainable Development Summit in September 2015, which adopted "Transforming our World: The 2030 Agenda for Sustainable Development" and codified the Sustainable Development Goals (SDGs) agreed upon by all member states. Taking into account the three dimensions of sustainable development (economic, social, and environmental), the Sustainable Development Goals (SDGs) comprise 17 primary goals and 169 secondary goals. For instance, promoting sustained, inclusive, and sustainable economic growth, inclusive and sustainable industrialization and innovation, addressing the climate crisis caused by greenhouse gas emissions, protecting, restoring, and promoting sustainable ecosystems, and reducing energy consumption and adopting affordable and clean energy.
China, as a key player in the sustainable development of the global economy, society, and environment, strives to practice green development while simultaneously accelerating its economic growth.
- Do public agencies control companies regarding the rules to be environmentally friendly? How?
Reply: According to the reviewer’s comments, we have added the contents as follows: (p5)
First, the government has boosted infrastructure construction through environmental subsidies in order to provide firms with adequate hardware support facilities for green innovation and other investment activities. This decreases the market risk associated with green innovation and increases the predictability of green technology product returns. Second, the local government's environmental expenditure communicates to the market the government's determination to carry out ecological and environmental protection and stimulates economic entities to improve environmental protection awareness, which on the one hand is conducive to directing social capital and high-quality talents to green fields such as ecology, energy conservation, environmental protection, and environmental governance, and on the other hand is conducive to fostering environmental governance. This will contribute to the expansion of green technology innovation. Thirdly, the government encourages businesses to adopt green investing practices in order to lower tax burdens by implementing environmental regulatory measures such as environmental levies.
- Introduction
- Raw 161, explain green technology spillovers. What kind of effects do companies have?
Reply: According to the reviewer’s comments, we have added the contents as follows: (p5)
That's because environmental subsidies will stimulate businesses to cut production costs and boost productivity through breakthroughs in green technology, which will directly raise their profit margins and entice additional businesses to follow suit [39]. Continuous innovation and widespread adoption of green technologies enable sustainable economic and environmental growth.
- Results
- .Is it possible to build extreme groups – calculate to upper and lower percentile of 20%? Your sample size is huge.
Reply: According to the reviewer’s comments, we have build extreme groups calculating to upper and lower percentile of 20%, and the results are the same as the results of our study. The results are as follows;
|
|
(upper percentile of 20%) |
( lower percentile of 20%) |
|
|
SGR |
SGR |
|
GI |
1.3817*** |
0.5893** |
|
|
(3.56) |
(2.24) |
|
SIZE |
-0.0028 |
0.0459*** |
|
|
(-0.21) |
(4.98) |
|
LEV |
0.2227*** |
0.0855*** |
|
|
(5.19) |
(3.29) |
|
ROA |
1.0965*** |
0.5145*** |
|
|
(9.99) |
(4.72) |
|
ATO |
-0.0193*** |
-0.0255*** |
|
|
(-4.54) |
(-4.04) |
|
GROWTH |
0.0503*** |
0.0197*** |
|
|
(4.07) |
(3.30) |
|
TOBIN |
-0.0061 |
0.0001 |
|
|
(-1.33) |
(0.06) |
|
AGE |
0.0567* |
-0.0239 |
|
|
(1.73) |
(-1.01) |
|
CFLOW |
-0.3530*** |
-0.0150 |
|
|
(-6.82) |
(-0.86) |
|
SOE |
0.0360* |
0.0029 |
|
|
(1.74) |
(0.25) |
|
_cons |
-0.2449 |
-0.9239*** |
|
|
(-0.72) |
(-4.70) |
|
Firm |
Yes |
Yes |
|
Industry |
Yes |
Yes |
|
Year |
Yes |
Yes |
|
N |
2956 |
2956 |
|
R2 |
0.183 |
0.235 |
|
adj. R2 |
0.178 |
0.230 |
|
F |
10.3606 |
8.3138 |
Notes: t-statistics in parentheses; *** p < 0.01, ** p < 0.05, *p < 0.1.
- .Regarding on the sample size, is it possible to split into different group? Are there different between males and females? Are there differences between the companies exist? à e.g. mining, textile, petroleum, chemical etc. are there any differences between them?
Reply: Because it is difficult to collect the sex data of the top manager, we have not split into males and females. The classification of industries are connected with the classification of polluting/non-heavily polluted industry, the related contents are as follows: (p21)
In this study we do not group the sample size in the basic regression and moderating effect regression analyses, focusing on reducing sample specificity as much as possible in order to empirically analyze the impact of green investment behavior on the sustainable development of listed enterprises in Shanghai and Shenzhen A-shares. Considering that enterprises may affect the regression results due to the different industries they belong to, the total sample size of 14,779 enterprises participating in the regression is divided into highly polluting enterprises (5,381) and non-highly polluting enterprises (9,398) in the further research analysis in Section 4.4. And the division of the sample is based on the "Guidelines on Environmental Information Disclosure of Listed Companies" issued by the Chinese Ministry of Environmental Protection in 2010. The "Guidelines" classify mining, textiles, paper, paper products, petroleum, chemicals, chemical fiber, ferrous metals, and aluminum as highly polluting industries, and other industries as non-highly polluting industries, according to the industry distinction. On this basis, the group regressions were conducted with corporate lagged one-period green investment as the independent variable and corporate sustainability as the dependent variable, and the results in Table 10 were obtained.
- How did Corona affect the results? Past – present – are there any differences?
Reply: We have tested the samples including the year of 2020 which is affected by the Corona and the outcome is consistent with the results of the previous regression analysis. The related contents are as follows: (p16)
Table 6 shows the results of the regression analysis of green investment for corporate sustainable development after removing the sample of companies from 2020. The regression coefficient of green investment for corporate sustainable development is 1.738 and thus passes the significance test, as shown in Column (1) of the main model results in Table 6. This outcome is consistent with the results of the previous regression analysis and suggests that green investment has a significant promotion effect on corporate sustainable development. After including the moderating variables in Columns (2)-(4), the regression results show that the coefficients of the three interaction terms are 0.013, 0.008, and 0.012 and that all three are statistically significant. When the moderating factors are included, the corrected R2 value increases dramatically. Consistent with the findings of prior regression analyses, the results of this study suggest that government environmental subsidies, investor attention, and executives' overseas experience positively moderate the impact of green investment on business sustainability
- Row 567-69 how do you choose both groups? Sample size of both groups? Even so Table 9 what is the variable to split into polluting/non-heavily polluted industry?
Reply: According to the reviewer’s comments, we have revised the contents as follows: (p21)
In this study we do not group the sample size in the basic regression and moderating effect regression analyses, focusing on reducing sample specificity as much as possible in order to empirically analyze the impact of green investment behavior on the sustainable development of listed enterprises in Shanghai and Shenzhen A-shares. Considering that enterprises may affect the regression results due to the different industries they belong to, the total sample size of 14,779 enterprises participating in the regression is divided into highly polluting enterprises (5,381) and non-highly polluting enterprises (9,398) in the further research analysis in Section 4.4. And the division of the sample is based on the "Guidelines on Environmental Information Disclosure of Listed Companies" issued by the Chinese Ministry of Environmental Protection in 2010. The "Guidelines" classify mining, textiles, paper, paper products, petroleum, chemicals, chemical fiber, ferrous metals, and aluminum as highly polluting industries, and other industries as non-highly polluting industries, according to the industry distinction. On this basis, the group regressions were conducted with corporate lagged one-period green investment as the independent variable and corporate sustainability as the dependent variable, and the results in Table 10 were obtained.
- Do you include a control group or is it possible to supply?
Reply: The purpose of this study is to investigate the characteristics of all listed companies in China and not to find the differences between groups. Also, it is difficult to collect the data of a control group. We hope we could investigate the differences between groups in the future.
- 10years – I miss a overview like a table of year and sample size
|
2010 |
2011 |
2012 |
… |
2020 |
|
N=XX |
|
|
|
|
à separating the industry, highlighting effects like the pandemic. Over the years, are the participants the same?
Reply: According to the reviewer’s comments, we have added the contents as follows: (p11-p12)
From Table 2, it can be seen that we screened a total sample size of 14,779 for 10 years, and the specific distribution of samples participating in the statistics for each year is shown in Table 2. Due to the one-period data that was used for the independent variables, there was no real sample size for the regression in 2010.
Table 2. Results of the Annual sample size distribution.
|
Year |
2010 |
2011 |
2012 |
2013 |
2014 |
2015 |
2016 |
2017 |
2018 |
2019 |
2020 |
|
Sample |
N=0 |
N=999 |
N=1157 |
N=1294 |
N=1337 |
N=1236 |
N=1333 |
N=1586 |
N=1764 |
N=2016 |
N=2057 |
- Discussion
- Overall, I miss some links to other studies relating to your findings.
Reply: According to the reviewer’s comments, we have added the contents as follows: (p22)
With the deterioration of the global ecological environment, the level of green investment and sustainability of companies, as important players contributing to socioeconomic development and environmental governance, are receiving increasing attention. scholars such as Balcilar argue that green energy consumption and investment have a small but positive impact on economic growth, and their study concludes that the capacity utilization of green energy consumption and investment has not yet developed to a level that can mitigate greenhouse effect and stimulate sustainable development in the long term to a viable level [87]. However, some scholars have argued that green investment has a key role in promoting high-quality economic development [17] and reducing carbon emissions [18]. In addition, many scholars also focus on the study of the relationship between environmental investment and financial performance and environmental performance, for example, Jin et al. argue that there is a u-shaped relationship between environmental investment and financial performance [88]; Shabbir et al. argue that there is a significant positive relationship between intra-firm environmental investment and firm financial performance [89]. Li et al. argue that there is a positive correlation [90], etc.

Reviewer 3 Report
Title: Can Enterprises in China Achieve Sustainable Development 2 Through Green Investment?
It is a nice paper indeed. In this study, authors claim that this study uses fixed effects regression models to analyse the impact of green investment on corporate sustainability in Chinese listed companies for the period of 2010 to 2020. It also investigates the moderating effects of government environmental subsidies, investor attention, and executives’ overseas experience on 14 the relationship between green investment and corporate sustainability. I believe it will be a good candidate for the publication at this journal. I think the following comments may help the authors to revise it accordingly”
1) Authors have portrayed the overall summary of the study in the abstract. It is easy to read and understand the main theme by going through the abstract. After reading the paper, it seems to me that authors follow quantitative approach that should be mention
in the abstract. It will be nice if authors mention the sampling technique for collecting the data from the sample companies. Also, need to mention the data analysis tools name here.
2) Authors nicely discussed the overall study theme, significance, objectives and past studies in relation to this study in the “1. Introduction” section. I strongly recommend the authors, to add contributions of study to the current literature in one or two paragraphs. Instead, you can remove the study finding in the Introduction section.
3) “2. Theoretical Background and Hypotheses Testing” is ok. Authors put the relevant information here to reflect the current study objective. It will be great if authors cite more relevant papers here in relation to the current study. The figure 1 is easy to understand.
4) In the “3. Research Methodology and Design:, authors mentioned that a total of 14,779 Chinese listed enterprises in the period of 2010–2020 served as the 274 source of data for this study. Unfortunately, authors did not mention the sample numbers among the total mentioned companies. Authors should explain the total data collection process. How did they collection? Which instruments did they use during data collection? Which tools did they used to analyse the data? How did they sort out the total samples from the large numbers? Sampling technique? Qualitative or quantitative? Authors should mention the instruments that they used for data collection in the abstract if they used any survey questionnaires?
5) “4. Results of the Empirical Analysis” is ok. Authors should arrange the tables nicely so that it looks attractive.
6) In “5. Conclusion”, authors discussed the contribution of this paper that is easy to understand. Authors should put a separate paragraph as a hint of “Implication of the study”. Also authors should separate the limitations of the study from the conclusion and put as a separate paragraph as a hint of “Limitations and Future Directions”.
7) I found errors in the references list that did not follow the journal’s referencing styles properly. Authors should check the referencing style in the authors’ guidelines and do it accordingly. Also, authors should check the in-text citations with references list, tables and figures numbers.
8) Authors should check the grammatical errors, typos and so on.
Author Response
Title: Can Enterprises in China Achieve Sustainable Development 2 Through Green Investment?
It is a nice paper indeed. In this study, authors claim that this study uses fixed effects regression models to analyse the impact of green investment on corporate sustainability in Chinese listed companies for the period of 2010 to 2020. It also investigates the moderating effects of government environmental subsidies, investor attention, and executives’ overseas experience on 14 the relationship between green investment and corporate sustainability. I believe it will be a good candidate for the publication at this journal. I think the following comments may help the authors to revise it accordingly”
1) Authors have portrayed the overall summary of the study in the abstract. It is easy to read and understand the main theme by going through the abstract. After reading the paper, it seems to me that authors follow quantitative approach that should be mention
in the abstract. It will be nice if authors mention the sampling technique for collecting the data from the sample companies. Also, need to mention the data analysis tools name here.
Reply: According to the reviewer’s comments, we have added the contents as follows: (p1)
This study uses fixed-effects regression models to analyze the impact of green investment on corporate sustainability in Chinese listed companies for the period of 2010 to 2020.
The data used in this study were not only from the China Stock Market & Accounting Research (CSMAR) database but also collected manually from the annual reports and social responsibility reports of listed companies using web crawler technology. And the robustness test was conducted by removing the epidemic year and replacing the range of independent variables and 2SLs. This study uses Stata 17.0 to filter and process the data.
2) Authors nicely discussed the overall study theme, significance, objectives and past studies in relation to this study in the “1. Introduction” section. I strongly recommend the authors, to add contributions of study to the current literature in one or two paragraphs. Instead, you can remove the study finding in the Introduction section.
Reply: According to the reviewer’s comments, we have added the contents as follows: (p2)
The Chinese government has used administrative regulations, economic instruments, and market mechanisms with some success to limit pollutant emissions [13], but if serious problems such as resource scarcity, environmental pollution, and ecological damage are not fundamentally addressed, China will struggle to achieve its green and sustainable development goals [14]. Environmental equity, according to Li and other scholars, is essential for achieving resource use efficiency and sustainable development, while local government environmental governance capacity and corporate green innovation technologies can effectively mitigate environmental inequality [15]. According to Ren et al., green investment can reduce environmental pollution by enhancing energy conservation and emission reduction capacities, expanding technological innovation capacities, and modernizing industrial structures [13]. Green energy consumption and investment have a small but positive impact on economic growth, according to Balcilar and other macroeconomists [16]. However, some academics are optimistic about the impact of green investment on economic development. The literature has evaluated the influence of green investment on high-quality economic development and carbon emission reduction, concluding that green investment has become crucial in promoting high-quality economic development [17] and reducing carbon emissions [18].
3) “2. Theoretical Background and Hypotheses Testing” is ok. Authors put the relevant information here to reflect the current study objective. It will be great if authors cite more relevant papers here in relation to the current study. The figure 1 is easy to understand.
Reply: According to the reviewer’s comments, we have added the contents as follows:
Studies based on neoclassical economic theory argue that corporate environmental protection expenditures crowd out productive capital, leading to a decrease in productivity and profit [25]. Green investment by businesses is a "passive activity" that complies with governments' environmental rules [26]. This characterization is due to the high barrier to entry for businesses looking to invest in environmental safeguards, such as the purchase of environmental protection equipment and funding of green technology research and development (R&D), and the low probability of a positive return [27]. According to studies based on the Porter hypothesis, environmental investments raise expenses in the short term but have the potential to spur technical innovation and generate "innovation compensation" over the long term, thereby improving productivity and performance [28]. Meanwhile, green investment has also been regarded as an "active behavior" of enterprises. On the one hand, businesses benefit from increasing the rate at which they use their resources, attracting more customers and other sources of funding, building a positive reputation among their target audiences, and achieving sustainable development [29]. Taylor et al. concluded that upholding environmental responsibility may result in a short-term loss of economic benefits but guarantees long-term sustainable development [30]. On the other hand, businesses can boost customer loyalty and reputation by engaging in green practices, thus reaping economic benefits from an indirect approach [31]. (p3-4)
According to Brown et al., government taxes encourage businesses that produce pollution to invest in adopting and using cleaner production methods. They contend that taxing pollutant emissions encourages businesses to invest more in green technologies to switch to cleaner production. Green investments, they claim, have higher marginal returns for polluting businesses [40] and are more likely to encourage long-term corporate growth. (p4)
Empirical evidence from Sun et al. supports the notion that green investments and clean energy both play significant roles in reducing environmental pollution and fostering environmentally sustainable development [43]. Xin et al. found that CSR can promote high-quality corporate development through improved green innovation, environmental investment, and corporate governance [44]. (p4)
Liu and other scholars concluded from empirical analysis that environmental taxes have a catalytic effect on firms to increase their environmental investments and that environmental taxes promote firm performance by increasing environmental investments [48]. (p5)
Using empirical analysis, Liu et al. came to the conclusion that environmental taxes improve firms' performance by spurring increased environmental investment and that environmental taxes have a catalytic effect on this increase in investment [48]. Government environmental spending, according to Hong et al., can lead to businesses adopting a free-rider mentality and avoiding environmental responsibility, which has a detrimental effect on how effective that spending is. Corporate green investment is negatively impacted by government environmental spending, or "crowding out" [56]. However, Ding et al. came to the conclusion that government subsidies have a favorable impact on corporate environmental investment [57] and that the association between government subsidies and corporate environmental investment can help decision-makers develop policies and allocate limited resources. (p5)
Deng et al. discovered that the occurrence of environmental events may strengthen firms' environmental investments and that extremely high investor attention leads to higher returns for green firms [59]. This, in turn, may lead firms to increase their green investments because of the attention that investors pay to green companies. (p6)
Othar Kordsachia and colleagues argued that the propensity for sustainability held by institutional investors is also a driving force behind firms' green investments. They argue that companies with more sustainable institutional investor ownership have higher carbon risk awareness and that sustainable institutional ownership is positively related to firms' environmental performance [64]. In addition, they claim that firms with more sustainable institutional investor ownership have higher environmental performance. According to the findings of Pan et al.'s research, public concern for the environment has a positive impact on the effectiveness of green investments made by businesses [65]. (p7)
Cao et al. argued that executives with overseas experience have substantial beneficial effects on the behavioral decisions of companies' green investment and environmental social responsibility [67], and how to make use of executives with overseas backgrounds to improve the efficiency of enterprises' green investment becomes an important issue. Cui et al. discovered that, when comparing the roles of executives with overseas backgrounds in high and low carbon firms, executives with overseas experience play a larger role in green innovation in high carbon firms and are more conducive to meeting carbon reduction goals [68] in response to social environment needs [69]. (p7)
Zhang et al. concluded that the overseas backgrounds of CEO-led companies provide higher quality sustainability reports [71], suggesting that executives with overseas backgrounds have higher ethics, integrity, and focus on sustainability. (p7)
Chen et al. concluded that the overseas background of executives has a significant positive impact on the green technology innovation of enterprises, and even its effect on the green innovation of enterprises is more obvious in high pollution cities [79]. (p8)
4) In the “3. Research Methodology and Design:, authors mentioned that a total of 14,779 Chinese listed enterprises in the period of 2010–2020 served as the 274 source of data for this study. Unfortunately, authors did not mention the sample numbers among the total mentioned companies. Authors should explain the total data collection process. How did they collection? Which instruments did they use during data collection? Which tools did they used to analyse the data? How did they sort out the total samples from the large numbers? Sampling technique? Qualitative or quantitative? Authors should mention the instruments that they used for data collection in the abstract if they used any survey questionnaires?
Reply: According to the reviewer’s comments, we have added the contents as follows: (p8-p9)
The Guide to Environmental Information Disclosure of Listed Companies, published by the former Ministry of Environmental Protection in 2010, proposed to further improve the transparency of corporate environmental information disclosure. On the one hand, considering the Chinese government's further emphasis on Chinese companies' participation in environmental protection after 2010, and on the other hand, considering the poor availability of data on corporate green investment before 2010, we set the sample period as 2010–2020. The authors firstly selected all listed companies in the A-share market of the Shanghai and Shenzhen Stock Exchanges from 2010 to 2020 through the China Stock Market & Accounting Research (CSMAR) database as the research sample. Second, enterprises with special treatment, special treatment*, or special transfer status were removed from the sample. Third, the samples that lacked critical information were discarded. Fourth, because the lagged period data are used for the independent variables in this study, the effective sample size is 14,779 after excluding the final sample that did not actually participate in the regression; Fifth, to reduce the effect of outliers, the variables were winsorized at the upper and lower 1% levels (excluding the dummy variables); sixth, we logged the continuous variables to reduce the interference of heteroskedasticity. This study uses Stata 17.0 to filter and process the data, and the results of the statistical analysis were obtained using Stata 17.0.
The data on green investment of listed companies in Shanghai and Shenzhen A-shares in China from 2010-2020 were obtained mainly by downloading annual reports of enterprises, information from the websites of listed companies, social responsibility reports of listed companies, and manually collating capital expenditure and expensed expenditure of enterprises on environmental protection and summing them up by using web crawler technology. The information on government funding for environmental protection was compiled from various sources, including annual company reports, social responsibility reports, company websites, and environmental department websites, using web crawling techniques and manually collating the aggregated results. The investor focus indicators were taken from the Google Search Volume Index (GSVI). Executives' overseas experience was gleaned from the biographical information in the CSMAR database, which, if necessary, was supplemented with the manually collated biographies of executives disclosed in annual reports. All other required public company data was obtained from the CSMAR database.
.
5) “4. Results of the Empirical Analysis” is ok. Authors should arrange the tables nicely so that it looks attractive.
Reply: We will arrange the tables nicely after accepting.
6) In “5. Conclusion”, authors discussed the contribution of this paper that is easy to understand. Authors should put a separate paragraph as a hint of “Implication of the study”. Also authors should separate the limitations of the study from the conclusion and put as a separate paragraph as a hint of “Limitations and Future Directions”.
Reply: According to the reviewer’s comments, we have revised the contents as follows: (p23)
- Implication of the study
The findings of our study offer several theoretical and practical implications for advancing green investment and encouraging sustainable business growth.
(1) Theoretically, the results widen the linked theories of green development and environmental governance and deepen the linkage between green investment and sustainable development at the micro level.
(2) In a practical sense, the results emphasize that businesses must first actively engage in green practices, uphold green management ideals, and enhance their green image so that green investment strategies can become an integral part of their long-term strategies and help them achieve sustainable development. Meanwhile, the government should increase environmental protection subsidies and formulate and adjust environmental regulation strategies in a reasonable and adaptable manner. On the one hand, such strategies can reduce the cost of green investment by enterprises, stabilize the normal operation of enterprises, ease financing constraints, and encourage enterprises to actively make green investments. On the other hand, the government's publication of the green signal "compels" businesses to effectively conduct green investments. In addition, relevant government departments should introduce correspondingly favorable policies to further expand the scale of institutional investors. We should also use big data to grasp investor opinion in a timely manner, consider the impact of investor attention on the sustainable development of enterprises, direct investors toward clean energy, renewable resources, and other green projects, and pay attention to CSR. Finally, the role of globalization and the risk aversion of executives with overseas experience in environmental awareness should be stressed, and corporate leaders should be directed to consider sustainable development alongside short-term economic gains while making decisions.
(3) Besides, it facilitates the designation of appropriate policies by regulators, and helps to develop some strategies and guidelines for emerging markets. First, the central government should strengthen environmental monitoring and implement differentiated incentives for green investments, as well as increase support for green technology innovation in manufacturing and private businesses, adapt taxation and other regulatory policies flexibly, and use emissions taxes to encourage high-polluting companies to invest in clean energy technologies. Second, managers and regulators should be aware of non-financial shareholder activism by investors, such as institutional investors' voting on compensation and other monitoring activities, and fully utilize institutional investors' role in keeping an eye on and directing businesses toward making green investments. Third, further optimize the strategy of introducing outstanding talents with overseas experience and attracting executives with overseas experience to participate in the core management of domestic enterprises through preferential welfare policies and compensation management, among other strategies, to improve the efficiency of green investment and the market competitiveness of domestic enterprises with their advanced environmental awareness and environmental vision accumulated overseas, thus driving the enterprises to achieve the goal of sustainable development.
5.4 Limitations and Future Directions
This study has some limitations with regard to the moderating effect found. Specifically, only three variables from typical stakeholder perspectives are selected to study the mechanism of green investment's effect on corporate sustainability. On the basis of the previous literature, we learn that other micro- and macro-level factors influence corporate sustainable development; these factors include the degree of regional economic development, external financing restrictions, internal enterprise control, and the degree of enterprises' green innovation. These factors are not fully covered in this study, but they can be explored more thoroughly and extensively in subsequent research.
In addition, the industry heterogeneity test reveals that heavily polluting industries are more effective at promoting sustainable development through green investment. In the next stage, more detailed research can be conducted on these industries to facilitate the provision of targeted and realistic guidance strategies for enterprises and to promote the overall green and sustainable development of society.
7) I found errors in the references list that did not follow the journal’s referencing styles properly. Authors should check the referencing style in the authors’ guidelines and do it accordingly. Also, authors should check the in-text citations with references list, tables and figures numbers.
Reply: According to the reviewer’s comments, we have checked referencing style and the in-text citations with references list, tables and figures numbers.
8) Authors should check the grammatical errors, typos and so on.
Reply: According to the reviewer’s comments, we have checked the grammatical errors, typos.

Round 2
Reviewer 1 Report
The authors done respective changes, so i recommend this article for publications. Congradulation to authors.